# Universally Amplifying Randomized Smoothing for Certified Robustness with Anisotropic Noise

## Abstract

Randomized smoothing has achieved great success for certified adversarial robustness. However, existing methods (especially the theory for certification guarantee) rely on a fixed i.i.d. noise distribution for all dimensions of the data (e.g., all the pixels in an image), and may result in limited performance of certified robustness. To address this limitation, we propose UCAN: a novel technique that Universally amplifies randomized smoothing for Certified robustness with Anisotropic Noise. It can theoretically transform any randomized smoothing method with isotropic noise to ensure certified robustness based on different variants of anisotropic noise. The theories universally work for using different noise distributions against different $\ell_p$ perturbations. Furthermore, we also design a novel framework with three example noise parameter generators (NPGs) for customizing the anisotropic noise. Finally, experimental results demonstrate that UCAN significantly outperforms the state-of-the-art (SOTA) methods, e.g., the certified accuracy can be improved by up to $182.6\%$ at large certified radii on MNIST, CIFAR10, and ImageNet datasets.

## 1 Introduction

Deep learning (DL) models have been proven to be vulnerable to well-crafted adversarial perturbations (Goodfellow et al., 2015; Carlini & Wagner, 2017). To protect DL models against adversarial attacks, defense methods with certified robustness guarantees are desired. Recently, randomized smoothing methods (Lecuyer et al., 2019; Teng et al., 2020; Cohen et al., 2019) were proposed to provide efficient certified robustness to any classifier and become the state-of-the-art. By injecting noise into the training and inference phases, randomized smoothing turns any classifier into a smoothed classifier. Theoretical bounds were then derived based on the noise distributions used for randomized smoothing. For example, Cohen et al. (2019) derives a tight $\ell_2$ certified radius for adopting the Gaussian noise to smoothen the classifier where the same distribution is used for all the data dimensions (called "isotropic noise"). More recently, some works (Zhang et al., 2020; Yang et al., 2020; Hong et al., 2022) improve the certified robustness performance of randomized smoothing by seeking better isotropic noise distributions, e.g., hyperbolic secant distribution (Hong et al., 2022), Pareto distribution (Yang et al., 2020), or general exponential distribution (Zhang et al., 2020).

However, most existing randomized smoothing methods cannot provide the best possible certification since their theories for certified robustness guarantees are limited to isotropic noise distributions. To unlock the certification guarantee with anisotropic noise-based randomized smoothing, there are two major challenges. The first challenge is how to develop novel theories *universally* for anisotropic noise distributions, in which noises with different means and variances are assigned to different data dimensions. The second challenge is how to assign proper means and variances for different data dimensions to optimize the certification performance.

In this paper, we propose a novel technique to *Universally amplify randomized smoothing for Certified robustness with Anisotropic Noise* ("UCAN"). Specifically, we first propose a *universal theory* that can convert any randomized smoothing-based certification with isotropic noise into certification with anisotropic noise (see Table 1). Second, we propose three different methods under a unified framework to customize the anisotropic noise distributions for different data dimensions. Thus, UCAN can

universally amplify the certification performance of all the existing randomized smoothing methods. In summary, UCAN makes the following key contributions to certified robustness:

1) **First Universal Theory for certification with Anisotropic Noise**. To our best knowledge, we take the first step to propose a universal theory for certifying the robustness of randomized smoothing with any (anisotropic) noise distribution. This new theory universally supports all the existing (and future) randomized smoothing methods using anisotropic noises for certification and against various $\ell_p$ perturbations (e.g., $\ell_1$, $\ell_2$,..., $\ell_\infty$).

2) **Novel Noise Parameter Generators (NPGs) for Customizing Anisotropic Noise**. We also design three NPGs (including two novel neural networks) to efficiently customize the element-wise hyper-parameters (mean and variance) in the anisotropic noise distributions for all the data dimensions. They significantly amplify the certification from different aspects.

3) **Significantly Boosted Certification**. Experimental results on benchmark datasets demonstrate that UCAN drastically outperforms the SOTA randomized smoothing-based certified robustness methods. For instance, the certified accuracy can be improved by $142.5\%$, $182.6\%$, and $121.1\%$ over the SOTAs on MNIST, CIFAR10, and ImageNet, respectively.

## 2 RELATED WORK

**Randomized Smoothing**. It was first studied by Lecuyer et al. (2019) based on the Differential Privacy theory Dwork (2006). Simultaneously, the first tight guarantee was proposed by Cohen et al. (2019), in which, the smoothed classifier's prediction (via Gaussian noise) can be tightly guaranteed to be consistent within a $\ell_2$ certified radius. Later, a series of methods have been proposed to guarantee the robustness against different $\ell_p$ perturbations with different noise distributions, e.g., Teng et al. (2020) derived the certified radius for $\ell_1$ perturbations with Laplace noise, and Lee et al. (2019) derived the certified radius against $\ell_0$ perturbations with uniform noise. Another line of methods (Zhang et al., 2020; Yang et al., 2020; Hong et al., 2022) proposed unified theories to guarantee the robustness against a diverse set of $\ell_p$ perturbations with different noises. However, all these methods (especially the theories) are limited to adopting isotropic noises for randomized smoothing.

**Data-Dependent Randomized Smoothing**. It aims to improve the certified robustness by optimizing the noise distribution for different inputs (*but still based on isotropic noise distributions due to the lack of theories for randomized smoothing with anisotropic noise*). For example, Alfarra et al. (2020) optimized the variance parameter in Gaussian distribution via the gradient of the certified radius w.r.t. the variance. Súkeník et al. (2021) considered the variance as a function of the input, and models the relationship between them. Wang et al. (2020) selected a proper variance by grid-search.

**Anisotropic Randomized Smoothing**. Recently Eiras et al. (2022) proposed a theorem for anisotropic randomized smoothing based on Lipschitz theories under a specific setting. However, its theory is based on assumptions that the networks are $L$-Lipschitz continuous and thus the universality is relatively limited. Furthermore, UCAN significantly outperforms Eiras et al. (2022) as shown in our experiments.

## 3 UCAN: THEOREM AND METRIC

### 3.1 RANDOMIZED SMOOTHING WITH ISOTROPIC NOISE FOR CERTIFIED ROBUSTNESS

We study the classification problem that maps input from $\mathbb{R}^d$ to classes $\mathcal{C}$. Given any base classifier $f$, existing randomized smoothing turns $f$ into a "smoothed" classifier $g$ with the isotropic noise. Given the noise $\epsilon \in \mathbb{R}^d$ from any isotropic distribution $\phi$, and let $X = x + \epsilon$, the smoothed classifier can be defined as: $g(x) = \arg\max_{c \in \mathcal{C}} \mathbb{P}(f(X) = c)$. Then existing theorems can be summarized as:

**Theorem 3.1** (**Certified Robustness via Randomized Smoothing with Isotropic Noise**). *Given a smoothed classifier $g$ based on arbitrary standard classifier $f$. For a specific $x \in \mathbb{R}^d$, let $X = x + \epsilon$, if the top-1 class is $c_A \in \mathcal{C}$, the lower bound probability of the top-1 classes $\underline{p_A} \in [0,1]$, and the upper bound probability of other classes $\overline{p_B} \in [0,1]$ satisfies:*

$$\mathbb{P}(f(X) = c_A) \geq \underline{p_A} \geq \overline{p_B} \geq \max_{c \neq c_A} \mathbb{P}(f(X) = c) \tag{1}$$

Table 1: Certified radii (binary-case) for randomized smoothing with isotropic and anisotropic noise. $d$ is the dimension size. $\Phi^{-1}$ denotes the inverse CDF of normal distribution. $\lambda$ is the scalar parameter of the isotropic noise. $\sigma_i$ is the multiplier for the scale parameter $\lambda$ in the $i$-th dimension

| Distribution | PDF | Adv. | Isotropic Guarantee | $\ell_p$ Radius for Iso. RS | Anisotropic Guarantee | $\ell_p$ Radius for Ani. RS | Alt. Lebesgue Measure |
|---|---|---|---|---|---|---|---|
| Gaussian Cohen et al. (2019) | $\propto e^{-\|\frac{x}{\lambda}\|_2^2}$ | $\ell_2$ | $\|\delta\|_2 \le \lambda(\Phi^{-1}(\underline{p_A}))$ | $\lambda(\Phi^{-1}(\underline{p_A}))$ | $\|\frac{\delta}{\sigma_i}\|_2 \le \lambda(\Phi^{-1}(\underline{p_A}'))$ | $\min\{\sigma_i\}\lambda(\Phi^{-1}(\underline{p_A}))$ | $\sqrt[d]{\prod_{i=1}^d \sigma_i}\lambda(\Phi^{-1}(\underline{p_A}'))$ |
| Gaussian Yang et al. (2020) | $\propto e^{-\|\frac{x}{\lambda}\|_2^2}$ | $\ell_1$ | $\|\delta\|_1 \le \lambda(\Phi^{-1}(\underline{p_A}))$ | $\lambda(\Phi^{-1}(\underline{p_A}))$ | $\|\frac{\delta'}{\sigma_i}\|_1 \le \lambda(\Phi^{-1}(\underline{p_A}'))$ | $\min\{\sigma_i\}\lambda(\Phi^{-1}(\underline{p_A}))$ | $\sqrt[d]{\prod_{i=1}^d \sigma_i}\lambda(\Phi^{-1}(\underline{p_A}'))$ |
|  |  | $\ell_\infty$ | $\|\delta\|_\infty \le \lambda(\Phi^{-1}(\underline{p_A}))/\sqrt{d}$ | $\lambda(\Phi^{-1}(\underline{p_A}))/\sqrt{d}$ | $\|\frac{\delta'}{\sigma_i}\|_\infty \le \lambda(\Phi^{-1}(\underline{p_A}'))/\sqrt{d}$ | $\min\{\sigma_i\}\lambda(\Phi^{-1}(\underline{p_A}))/\sqrt{d}$ | $\sqrt[d]{\prod_{i=1}^d \sigma_i}\lambda(\Phi^{-1}(\underline{p_A}'))/\sqrt{d}$ |
| Laplace Teng et al. (2020) | $\propto e^{-\|\frac{x}{\lambda}\|_1}$ | $\ell_1$ | $\|\delta\|_1 \le -\lambda\log(2(1-\underline{p_A}))$ | $-\lambda\log(2(1-\underline{p_A}))$ | $\|\frac{\delta'}{\sigma_i}\|_1 \le -\lambda\log(2(1-\underline{p_A}'))$ | $-\min\{\sigma_i\}\lambda\log(2(1-\underline{p_A}'))$ | $\sqrt[d]{\prod_{i=1}^d \sigma_i}\lambda\log(2(1-\underline{p_A}'))$ |
| Exp. $\ell_\infty$ Yang et al. (2020) | $\propto e^{-\|\frac{x}{\lambda}\|_\infty}$ | $\ell_1$ | $\|\delta\|_1 \le 2d\lambda(\underline{p_A}-\frac12)$ | $2d\lambda(\underline{p_A}-\frac12)$ | $\|\frac{\delta'}{\sigma_i}\|_1 \le 2d\lambda(\underline{p_A}'-\frac12)$ | $2\min\{\sigma_i\}d\lambda(\underline{p_A}'-\frac12)$ | $2\sqrt[d]{\prod_{i=1}^d \sigma_i}d\lambda(\underline{p_A}'-\frac12)$ |
|  |  | $\ell_\infty$ | $\|\delta\|_\infty \le \lambda\log(\frac{1}{2(1-\underline{p_A})})$ | $\lambda\log(\frac{1}{2(1-\underline{p_A})})$ | $\|\frac{\delta'}{\sigma_i}\|_\infty \le \lambda\log(\frac{1}{2(1-\underline{p_A})})$ | $\min\{\sigma_i\}\lambda\log(\frac{1}{2(1-\underline{p_A}')})$ | $\sqrt[d]{\prod_{i=1}^d \sigma_i}\lambda\log(\frac{1}{2(1-\underline{p_A}')})$ |
| Uniform $\ell_\infty$ Lee et al. (2018) | $\propto \mathbb{I}(\|z\|_\infty \le \lambda)$ | $\ell_1$ | $\|\delta\|_1 \le 2\lambda(\underline{p_A}-\frac12)$ | $2\lambda(\underline{p_A}-\frac12)$ | $\|\frac{\delta'}{\sigma_i}\|_1 \le 2\lambda(\underline{p_A}'-\frac12)$ | $2\min\{\sigma_i\}\lambda(\underline{p_A}'-\frac12)$ | $2\sqrt[d]{\prod_{i=1}^d \sigma_i}\lambda(\underline{p_A}'-\frac12)$ |
|  |  | $\ell_\infty$ | $\|\delta\|_\infty \le 2\lambda(1-\sqrt[d]{\frac32-\underline{p_A}})$ | $2\lambda(1-\sqrt[d]{\frac32-\underline{p_A}})$ | $\|\frac{\delta'}{\sigma_i}\|_\infty \le 2\lambda(1-\sqrt[d]{\frac32-\underline{p_A}'})$ | $2\min\{\sigma_i\}\lambda(1-\sqrt[d]{\frac32-\underline{p_A}'})$ | $2\sqrt[d]{\prod_{i=1}^d \sigma_i}\lambda(1-\sqrt[d]{\frac32-\underline{p_A}'})$ |
| Power Law $\ell_\infty$ Yang et al. (2020) | $\propto \frac{1}{(1+\|\frac{z}{\lambda}\|_\infty)^a}$ | $\ell_1$ | $\|\delta\|_1 \le \frac{2d\lambda}{a-d}(\underline{p_A}-\frac12)$ | $\frac{2d\lambda}{a-d}(\underline{p_A}-\frac12)$ | $\|\frac{\delta'}{\sigma_i}\|_1 \le \frac{2d\lambda}{a-d}(\underline{p_A}'-\frac12)$ | $\min\{\sigma_i\}\frac{2d\lambda}{a-d}(\underline{p_A}'-\frac12)$ | $2\sqrt[d]{\prod_{i=1}^d \sigma_i}\frac{d\lambda}{a-d}(\underline{p_A}-\frac12)$ |

*then the prediction of $g$ on the perturbed input $x + \delta$ will consistently be $c_A$ if $\|\delta\|_p < R(\underline{p_A}, \overline{p_B})$, where $\|\cdot\|_p$ denotes the $\ell_p$-norm, and $R(\cdot)$ denotes a general function of certified radius formulas.*

The certified radius formula varies when the noise PDFs are different. We list several certified radius functions of existing randomized smoothing theorems *with isotropic noise* in Table 1 (left 5 columns). The Alternative Lebesgue Measure will be introduced as an anisotropic measure for anisotropic RS in Section 3.2.

## 3.2 THEOREM FOR ANISOTROPIC NOISE

In this section, we establish a universal theory for the certification via randomized smoothing with *anisotropic* noise. Given any isotropic randomized smoothing methods, our method can universally transform them to anisotropic randomized smoothing for certified robustness.

Specifically, given an arbitrary isotropic noise with zero-mean and the scale parameter $\lambda$ (w.r.t. the variance of the PDF) in each dimension, we can represent any anisotropic noise with the anisotropic mean offsets $\mu = [\mu_1, \mu_2, ..., \mu_d]$ and the anisotropic multipliers of the scale parameter $\sigma = diag(\sigma_1, \sigma_2, ..., \sigma_d)$ with $\sigma_i$ for the $i$-th dimension, and $\sigma_i > 0$. Then, the $i$-th dimension of the anisotropic noise has the mean $\mu_i$ and scale parameter $\sigma_i\lambda$, where $\lambda$ is multiplied by $\sigma_i$. Formally, denoting any isotropic noise as $\epsilon$ (generated by $\lambda$), we define the respective anisotropic noise $\epsilon'$ as:

$$\epsilon' = \epsilon^\top \sigma + \mu \tag{2}$$

Then robustness of randomized smoothing with anisotropic noise can be ensured per Theorem 3.2.

**Theorem 3.2** (**Anisotropic Randomized Smoothing via Universal Transformation**). *Let $f : \mathbb{R}^d \to \mathcal{C}$ be any deterministic or random function. Suppose that for the multivariate random variable with isotropic noise $X = x + \epsilon$ in Theorem 3.1, the certified radius function is $R(\cdot)$. Then, for the corresponding anisotropic input $Y = x + \epsilon^\top \sigma + \mu$, if there exist $c'_A \in \mathcal{C}$ and $\underline{p_A}', \overline{p_B}' \in [0,1]$ such that:*

$$\mathbb{P}(f(Y) = c'_A) \ge \underline{p_A}' \ge \overline{p_B}' \ge \max_{c \ne c'_A} \mathbb{P}(f(Y) = c) \tag{3}$$

*then $g'(x+\delta') \equiv \arg\max_{c \in \mathcal{C}} \mathbb{P}(f(Y + \delta') = c) = c'_A$ for all $\|\frac{\delta'_i}{\sigma_i}\|_p \equiv (\sum_i^d (\frac{\delta'_i}{\sigma_i})^p)^{\frac1p} \le R(\underline{p_A}', \overline{p_B}')$ where $g'$ denotes the smoothed classifier based on anisotropic noise, $\delta'$ the perturbation on $x$, and $i$ the dimension index.*

*Proof.* See the detailed proof in Appendix A. □

In Table 1, we derive the corresponding certified radius functions of anisotropic noise-based randomized smoothing methods, transformed from most of the existing randomized smoothing methods with isotropic noise (derived based on Theorem 3.2). For other randomized smoothing methods (e.g., Zhang et al. (2020); Hong et al. (2022)) without explicit certified radius functions, our transformation method can be directly applied to the numerical certified radius result. We also present the binary case of Theorem 3.2 (binary classifier) in Appendix B.

In Figure 1(a), we illustrate the benefits that the anisotropic noise can bring to randomized smoothing. In Theorem 3.2, we observe that the mean offset $\mu_i$ does not affect the derivation of the certified robustness with anisotropic noise. Thus, it is likely that the probabilities $\underline{p_A}'$ and $\overline{p_B}'$ can be

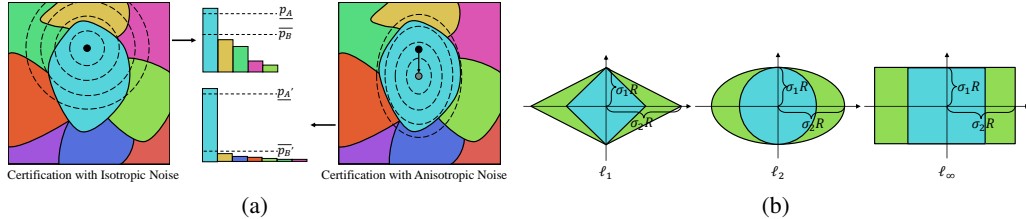

Figure 1: **(a)** The benefits of anisotropic noise over isotropic noise. When evaluating $x$ over smoothed classifiers, the decision regions of the base classifier $f$ are denoted in different colors. The dashed lines are the level sets of the noise distribution. The left figure shows the randomized smoothing with isotropic Gaussian noise $\mathcal{N}(0, \lambda^2 \mathbf{I})$ in Cohen et al. (2019) whereas the right figure illustrates the randomized smoothing with anisotropic Gaussian noise $\mathcal{N}(\mu, \Sigma)$, where $\Sigma = \lambda^2 diag(\sigma_1^2, \sigma_2^2, ..., \sigma_d^2)$. The anisotropic noise can improve the certified robustness by improving the gap of $\underline{p_A}$ and $\overline{p_B}$. **(b)** Illustration of the robustness region constructed with isotropic (blue) and anisotropic (green) noise.

improved by a proper mean offset in the anisotropic noise. Also, with the heterogeneous variance, the anisotropic noise can fit different dimensions of the input better without over-distortion.

**Boundary for Certified Region**. In Corollary 3.3, we derive the certified radii $R'$ for anisotropic randomized smoothing in the formation of isotropic $\ell_p$-ball, i.e., $||\delta'||_p \leq R'$. However, certified radii might be inaccurate for evaluating the certified robustness under anisotropic circumstances due to the asymmetric shape of the robustness region constructed by the anisotropic noise. In other words, the certified area represented by the isotropic $\ell_p$-ball is a subset of anisotropic certified region depicted by $(\sum_i^d (\frac{\delta_i'}{\sigma_i})^p)^{\frac{1}{p}} \leq R(\underline{p_A}', \overline{p_B}')$ (see Figure 1(b) for illustration).

**Corollary 3.3.** *For the anisotropic input $Y$ in Theorem 3.2, if the condition in Eq. (3) is satisfied, then $g'(x + \delta') \equiv \arg\max_{c \in \mathcal{C}} \mathbb{P}(f(Y + \delta') = c) = c_A'$ for all $||\delta'||_p \leq R'$ such that*

$$R' = \min\{\sigma_i\}R \tag{4}$$

*where $R$ is the corresponding certified radius of randomized smoothing via isotropic noise, and $\min\{\cdot\}$ denotes the minimum entry.*

*Proof.* The guarantee in Theorem 3.2 holds for $||\frac{\delta_i'}{\sigma_i}||_p \leq R$. Since $||\frac{\delta_i'}{\sigma_i}||_p \leq ||\frac{\delta_i'}{\min\{\sigma_i\}}||_p$, if $||\frac{\delta_i'}{\min\{\sigma_i\}}||_p \leq R$, the guarantee still holds. This requires $||\delta'||_p \leq \min\{\sigma_i\}R$. $\qquad\square$

Besides the $\ell_p$ radii, we also introduce a general metric, i.e., Alternative Lebesgue Measure (ALM) Eiras et al. (2022), which is defined as $ALM = \sqrt[d]{\prod_{i=1}^d \sigma_i} R$ for auxiliary evaluation on the anisotropic robustness region (note that $\ell_p$ radius fall short to evaluate the robustness gain in some dimensions due to its symmetry, see Figure 1(b)). ALM is equivalent to the certified radii when applied to traditional isotropic RS where $\sigma_i = 1$ for all dimensions. See Eiras et al. (2022) and Appendix C for detailed theoretical analysis and discussions about the ALM.

## 4 CUSTOMIZING ANISOTROPIC NOISE

With Theorem 3.2, it is feasible to assign heterogeneous noise parameters to different data dimensions. However, finding more optimal heterogeneous noise parameters rather than randomly assigning them remains a challenge. To this end, in UCAN, we design a unified framework to customize anisotropic noise for randomized smoothing (see Figure 2(a)), which integrates three *noise parameter generators* (NPGs) with different scales of trainable parameters (see Figure 2(b)) and optimality:

1) Customizing **pattern-fixed anisotropic noise** by assigning heterogeneous noise parameters based on certain patterns of the data. The NPG is *training-free* and *inference-free*, but with *low optimality*.

2) Customizing **universal anisotropic noise** by optimizing trainable noise parameter generator according to a specific dataset. It requires the *pre-training* of the NPG (a neural network) and *one-time inference for the dataset*, with *moderate optimality*.

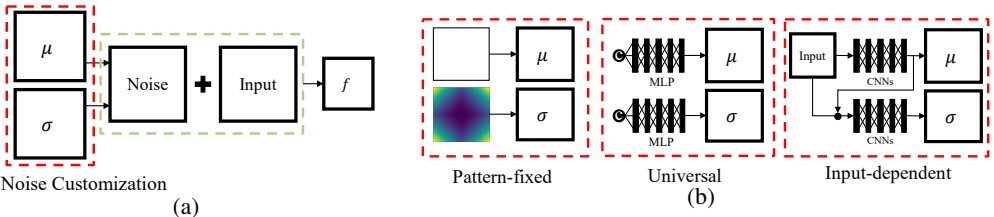

Figure 2: **(a)** The framework for customizing anisotropic noise for randomized smoothing. **(b)** Three example noise parameter generators (NPGs).

3) Customizing **input-dependent anisotropic noise** by optimizing the input-dependent noise parameter generator during training. It requires the *pre-training* of the NPG (a neural network) and *one-time inference for each input*, but with *high optimality*.

Note that these three different types of NPGs are only example methods. Other methods can also be designed with different objectives. The practical algorithms are detailed in Appendix F.

## 4.1 PATTERN-FIXED ANISOTROPIC NOISE

The NPG for pattern-fixed anisotropic noise is motivated by the intuitive understanding that different portion of the data (e.g., areas on the image) may influence the prediction variably (Gilpin et al., 2018). Typically, an image's center, where an object likely exists, may include more visual information than its borders, necessitating lower variance in the center to avoid blurring informative areas; while the borders can tolerate larger noise (e.g., higher variance) without notably hindering classification performance. Hence, this NPG assigns fixed spatial patterns for anisotropic noise.

Specifically, let the spatial distribution of the variances follow a function $\sigma(a, b)$, where $a$ and $b$ are the pixel's coordinates of the horizon and vertical axes such that the center of the image is denoted as $(a, b) = (0, 0)$. As discussed above, the central variance can be intuitively set to be smaller than the border's variance. Thus, we design three different types of spatial distribution as follows:



$\sigma(a,b) = \|(a,b)\|_1^2 + 1$    $\sigma(a,b) = \|(a,b)\|_2^2 + 1$    $\sigma(a,b) = \|(a,b)\|_\infty^2 + 1$

Figure 3: Spatial distributions for variances of the noise parameters (pattern-fixed)

$$\sigma(a, b) = \kappa \|(a, b)\|_p^2 + \iota, \quad p = 1, 2, \infty \quad (5)$$

where $\|(a, b)\|_p$ denotes the $\ell_p$-norm distance between $(a, b)$ and $(0, 0)$, $\kappa$ is a constant parameter tuning the overall magnitude of the variance, $\iota$ denotes the variance of a center pixel since $\sigma(0, 0) = \iota$, and $\iota > 0$ such that $\sqrt[d]{\prod_{(a,b)} \sigma(a, b)} \neq 0$. Figure 3 shows three examples of the spatial distribution when $p = 1, 2, \infty$, where $\kappa$ and $\iota$ are set as 1. The noise mean is set as 0 to avoid unnecessary deviation in the images.

The pattern-fixed anisotropic noise is an intuitive improvement by considering the different contributions to the classification results by different portions of the data (e.g., pixels). However, it is not sufficiently fine-tuned especially considering the diverse characteristics of different datasets. In the next subsection, we propose an automated approach that can derive the near-optimal spatial distribution of the variances for the anisotropic noise for each dataset.

## 4.2 UNIVERSAL ANISOTROPIC NOISE

This NPG leverages a fix-input neural network generator (Creswell et al., 2018) to learn anisotropic variances during the robust training (i.e., training with noise), which is shown in Figure 2(b). Specifically, the generator takes a fixed constant as the input and outputs an anisotropic $\sigma$ tensor and $\mu$ tensor with each element $\mu_i$ and $\sigma_i$ denoting the mean offset and the multiplier of noise (viz. variance) per pixel, respectively. Given any noise $\epsilon$ drawn from any noise distribution, the anisotropic noise can be generated as $\epsilon' = \epsilon^\top \sigma + \mu$. Then, the noise will be added to the input for randomized smoothing.

**Architecture**. We adapt the generator architecture (a multi-layer perception) in Generative Adversarial Network (GAN) (Goodfellow et al., 2020) to design a novel neural network generator. Different

from GAN, this NPG does not depend on the input data but depends on the entire dataset. Therefore, we fixed the input as constants. The NPG consists of $5$ linear layers and the first $4$ of them are followed by activation layers. The output will be transformed by a hyperbolic tangent function with an amplification factor $\gamma$, i.e., $\gamma tanh(\cdot)$. This amplified hyperbolic tangent layer is to limit the value of the variances since an infinite value in the noise parameters will fail the training.

**Loss Function**. To train the NPG towards a desired convergence, we need to design a proper loss function. Note that certified robustness with anisotropic noise can be measured by the ALM in a general way. Then, the NPG training should aim to maximize the product $\sqrt[d]{\prod \sigma}$ and the traditional certified radius $R$. To increase the product, we alternatively maximize the average $\sigma$ since maximizing the product could lead to unstable training. Since the certified radius is a function of $p_A$, improving the prediction accuracy over the noise can improve the certified region (measured by $\overline{\text{ALM}}$), which is also the goal of training the smoothed classifier. The loss function can be formally defined as:

$$\mathcal{L}(\theta_f, \theta_g) = \underbrace{-\frac{1}{d}\sum_i (\sigma_i(\theta_g))}_{\textit{Variance Loss}} \underbrace{- \sum_{k=1}^{N} y_k \log \hat{y}_k(x + \epsilon^\top \sigma(\theta_g) + \mu(\theta_g), \theta_f, \theta_g)}_{\textit{Smoothing Loss}}$$

where $\theta_f$ and $\theta_g$ denote the model parameters of the classifier and parameter generator, respectively, $k$ denotes the prediction class, $N$ represents the total number of classes, $y_k$ denotes the label of input $x$, and $\hat{y}_k$ is the prediction of $y_k$. The training of the NPG for $\mu$ is also guided by the smoothing loss to improve the prediction over the dataset.

**Universality**. The universality of noise and that of our robustness are different: the former focuses on using one noise to universally protect an entire dataset, and the latter focuses on defending against a wide range of perturbations (e.g., different $\ell_p$-norms) with various anisotropic noise distribution.

### 4.3 INPUT-DEPENDENT ANISOTROPIC NOISE

Although the universal anisotropic noise optimizes the noise parameters during training to provide better robustness guarantees, it is still not sufficiently fine-tuned to fully capture the heterogeneity between different input samples. Since the certification is also an input-dependent process that provides specific guarantees for different inputs (the guarantee is only valid on the corresponding certified input), it is intuitive to generate the best anisotropic noise for each data sample. Therefore, we design an input-dependent NPG to produce the anisotropic noise, which considers additional heterogeneity between different inputs besides the heterogeneity between the data dimensions.

Different from generating the universal anisotropic noise, *the parameter generators for mean and variance are not in parallel, but cascaded* (see Figure 2(b)). Specifically, in the training and certification, the mean parameter generator takes the input $x$ and returns a $\mu$ map. Then the variance parameter generator takes $x + \mu$ as the input to returns a $\sigma$ map (scale parameter of the noise). After that, the noisy input will be injected to the base classifier to train the smoothed classifier.

**Architecture**. This NPG learns the mapping from the image to the $\mu$ and $\sigma$ maps, which is similar to the function of neural networks in image transformation. Hence, inspired by the image super-resolution (Zhang et al., 2018), we also adapt the "dense blocks" Huang et al. (2017) as the main architecture to design the NPG. The details for the architecture can be found in Appendix E.

**Loss Functions**. The loss function is similar to that used for the universal anisotropic noise, but the NPG takes $x$ as input and outputs $\mu$ and $\sigma$.

## 5 EXPERIMENTS

We comprehensively evaluate UCAN in this section. Specifically, in Section 5.1, we test UCAN with the three different ways of generating anisotropic noises and compare them with the randomized smoothing baseline with isotropic noises. In Section 5.2, we thoroughly evaluate the universality of UCAN, including the universality on noise distributions and against different $\ell_p$ perturbations. In Section 5.3, we benchmark the best performance of UCAN with the SOTA RS methods.

**Metrics**. We derive the certified accuracy per the Alternative Lebesgue Measure (ALM) Eiras et al. (2022), which can be defined as the fraction of the test set that is certified to be consistently

correct *within the ALM (certified region)*. Formally, certified accuracy w.r.t. certified radius and ALM can be defined as: $Acc(R) = \frac{1}{N}\sum_{j=1}^{N} \mathbf{1}_{[g'(x^j+\delta)=y^j]}, \forall ||\delta||_p \leq R$ and $Acc(ALM) = \frac{1}{N}\sum_{j=1}^{N} \mathbf{1}_{[g'(x^j+\delta)=y^j]}, \forall ||\delta||_p \leq \sqrt[d]{\prod \sigma_i}R$, respectively, where $x^j$ and $y^j$ denote the $j$-th sample and its label in the test set, respectively. $N$ denotes the number of inputs/images in the test set.

To fairly position our methods, when compared to the SOTA methods (mostly isotropic), we present the certified accuracy w.r.t. both the $\ell_p$ *radius* and the ALM.

**Experimental Settings**. We test UCAN on three image datasets: MNIST (LeCun et al., 2010), CIFAR10, (Krizhevsky et al., 2009) and ImageNet (Russakovsky et al., 2015). Following Cohen et al. (2019) on certification, we obtain the certified accuracy on the entire test set in CIFAR10 and MNIST while randomly picking 500 samples in the test set of ImageNet; we set $\alpha = 0.001$ and the numbers of Monte Carlo samples $n_0 = 100$ and $n = 100,000$. More details about the experimental settings can be found in Appendix I.

### 5.1 COMPARISON OF RANDOMIZED SMOOTHING WITH ANISOTROPIC AND ISOTROPIC NOISE

We first evaluate randomized smoothing with anisotropic noise generated by the three example NPGs. W.l.o.g., we adopt Gaussian distribution (with zero-mean) to generate the anisotropic noise for randomized smoothing against $\ell_2$ perturbation and compare with the isotropic Gaussian baseline Cohen et al. (2019), which derives the tight certified radius (under multi-class setting) against $\ell_2$ perturbations. Other distributions against different $\ell_p$ perturbations (*universality* of UCAN) are detailed in Section 5.2.

**Parameter Setting**. We follow Cohen et al. (2019) to set different variances for isotropic Gaussian noise. For our pre-assigned anisotropic noise, since the variance varies in different dimensions, we re-scale $\sigma(a, b)$ such that $\sqrt[d]{\prod \sigma_i} = 1.0$. For the universal method, we set $\gamma = 5$ for MNIST and CIFAR10 and $\gamma = 2$ for ImageNet to achieve the best trade-off. After training, $\sqrt[d]{\prod \sigma_i}$ for MNIST, CIFAR10, and ImageNet are 1.56, 0.93, and 0.92, respectively. For the input-dependent method, the amplified factor $\gamma$ in the parameter generator is set as 1.0 for all the datasets.

**Experimental Results**. The experimental results are presented in Figure 4(a)-4(c). It shows that the certified accuracy of pattern-fixed anisotropic noise is strictly above the baseline with variance $\lambda = 1.0$ on all the datasets. Note that when $\sqrt[d]{\prod \sigma_i} = 1.0$, the Alternative Lebesgue Measure is equal to $R$. Therefore, it suggests that with the same level of variance, the anisotropic noise can achieve higher prediction accuracy since the key parts of the images (showing the objects) were less perturbed (see visualized examples in Figure 7(b)). Figure 4(d)-4(f) shows that the universal anisotropic noise significantly boosts the certified accuracy on CIFAR10 and achieves the best trade-off between certified accuracy and Alternative Lebesgue Measure (certified region). The certified accuracy is improved up to 39%. Figure 4(g)-4(i) shows that the input-dependent anisotropic noise significantly boosts the certified accuracy. The best improvement of the certified accuracy is 54%, and 35% for CIFAR10, and ImageNet, respectively, since the noise parameter generator (NPG) learns the best spatial distribution of noise parameters (see the examples in Figure 7(d)).

### 5.2 UNIVERSALITY (DIFFERENT NOISE PDFs AGAINST DIFFERENT $\ell_p$ PERTURBATIONS)

In this section, we evaluate the universality of UCAN over different noise distributions against different $\ell_p$ perturbations. Specifically, we evaluate the universality over noise distributions listed in Table 1. For a fair comparison, we follow Yang et al. (2020) to set the scalar parameter $\lambda$ of different noise distributions such that the variance is equal to 1. For the anisotropic noise, we follow the settings in Section 5.1 to set the parameters for pattern-fixed, universal, and input-dependent anisotropic noise. We only present the $\ell_2$ pattern of the pattern-fixed anisotropic noise due to the similar performance of different patterns. We only present the certified defenses against $\ell_1$ and $\ell_\infty$ perturbations due to lack of existing works against $\ell_2$ perturbations in Table 1 (the comparison with Cohen et al. (2019) against $\ell_2$ perturbations has been given earlier).

The experimental results are shown in Figure 5. In all settings, UCAN can universally amplify the certified robustness of randomized smoothing with isotropic noise.

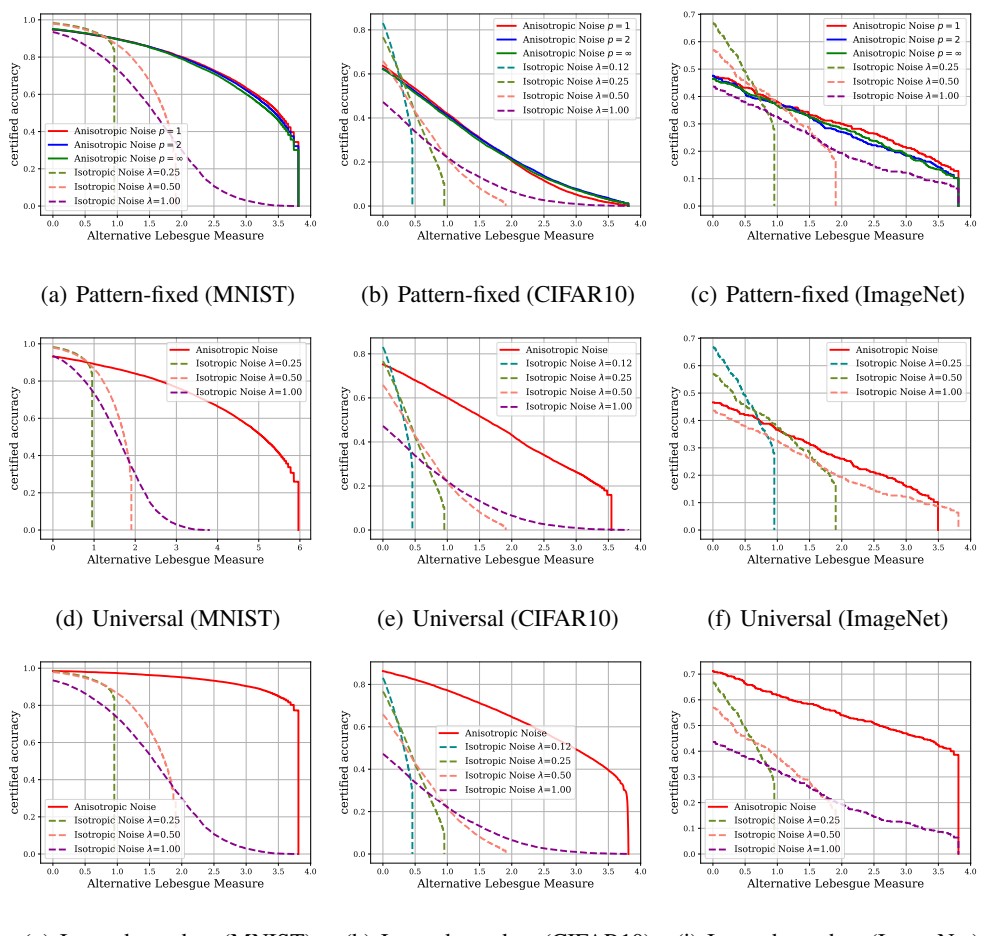

(a) Pattern-fixed (MNIST)    (b) Pattern-fixed (CIFAR10)    (c) Pattern-fixed (ImageNet)

(d) Universal (MNIST)    (e) Universal (CIFAR10)    (f) Universal (ImageNet)

(g) Input-dependent (MNIST)    (h) Input-dependent (CIFAR10)    (i) Input-dependent (ImageNet)

Figure 4: Comparison of randomized smoothing with anisotropic noise and that with isotropic noise (Gaussian distribution for certified defense against $\ell_2$ perturbations, comparing with Cohen et al. (2019)) – UCAN gives significantly better certified accuracy and larger certified region.

Table 2: Certified accuracy vs. $\ell_2$ perturbations (MNIST).

| Radius and ALM (equivalent for isotropic) | 0.0 | 0.25 | 0.50 | 0.75 | 1.00 | 1.25 | 1.50 | 1.75 | 2.00 | 2.25 |
|---|---|---|---|---|---|---|---|---|---|---|
| Cohen's Cohen et al. (2019) | 83% | 61% | 43% | 32% | 22% | 17% | 14% | 9% | 7% | 4% |
| Sample-wise Wang et al. (2020) | 98% | 97% | 96% | 93% | 88% | 81% | 73% | 57% | 41% | 25% |
| Input-depend Súkeník et al. (2021) | 99% | 98% | 97% | 94% | 88% | 79% | 58% | 27% | 0% | 0% |
| MACER Zhai et al. (2020) | 99% | 99% | 96% | 95% | 90% | 83% | 73% | 50% | 36% | 28% |
| SmoothMix Jeong et al. (2021) | 99% | 99% | 98% | 97% | 93% | 89% | 82% | 71% | 45% | 37% |
| DRT Yang et al. (2021) | 99% | 98% | 98% | 97% | 93% | 89% | 83% | 70% | 48% | 40% |
| Ours (certified accuracy w.r.t. radius) | 99% | 99% | 99% | 99% | 99% | 99% | 98% | 98% | 98% | 97% |
| Ours (certified accuracy w.r.t. ALM) | 98% | 98% | 98% | 98% | 97% | 97% | 96% | 96% | 95% | 94% |
| Improvement over Baseline (%) | +0.0% | +0.0% | +1.0% | +2.1% | +6.5% | +11.2% | +18.1% | +38.0% | +104.2% | +142.5% |

## 5.3 BEST PERFORMANCE COMPARISON VS. SOTA METHODS

We also compare our best performance (certification with input-dependent anisotropic noise) with the best performance of 12 SOTA methods.[1] Here we present the certified accuracy w.r.t. both *ALM* and $\ell_p$ *radius. Note that the ALM and radius are equivalent for SOTA methods with isotropic noise.*

Following the same settings in such existing randomized smoothing methods Cohen et al. (2019); Alfarra et al. (2020); Súkeník et al. (2021); Wang et al. (2020), we focus on the Gaussian noise against $\ell_2$ perturbations to benchmark with them. Results are shown in Table 2, 3, and 4. We mark the best performance of our methods and the baselines as red and blue, respectively. We also present the improvement of our method over the best baseline in percentage.

---

[1]W.l.o.g., we compare them on $\ell_2$ perturbations, and can draw similar observations on other $\ell_p$ perturbations.

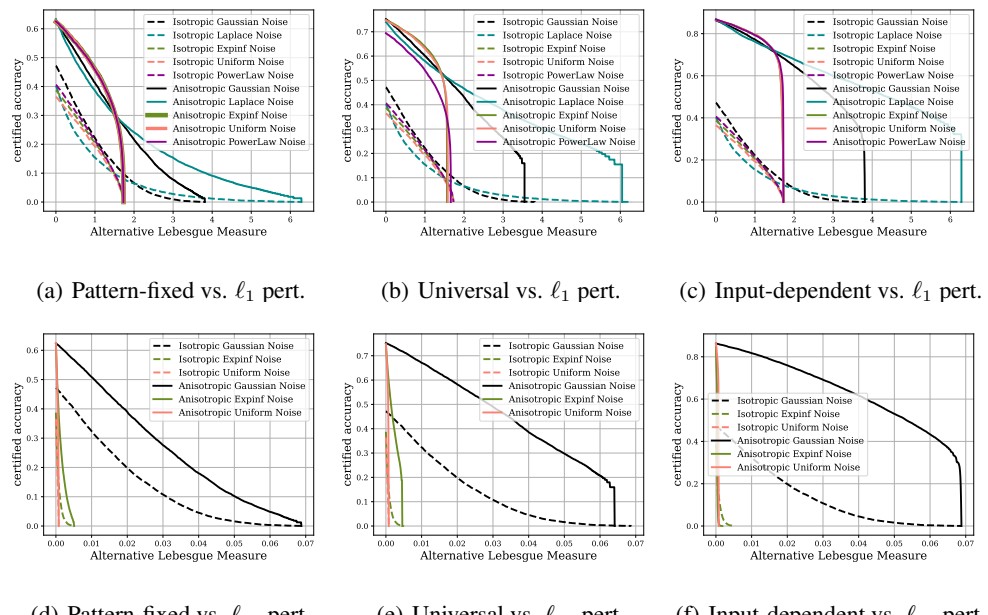

Figure 5: UCAN with three types of anisotropic noise (AN) vs. randomized smoothing with isotropic noise – different noise PDFs against different $\ell_p$ perturbations (universality) on CIFAR10.

Table 3: Certified accuracy vs. $\ell_2$ perturbations (CIFAR10).

| Radius and ALM (equivalent for isotropic) | 0.0 | 0.25 | 0.50 | 0.75 | 1.00 | 1.25 | 1.50 | 1.75 | 2.00 | 2.25 |
|---|---|---|---|---|---|---|---|---|---|---|
| Cohen's Cohen et al. (2019) | 83% | 61% | 43% | 32% | 22% | 17% | 14% | 9% | 7% | 4% |
| SmoothAdvSalman et al. (2019) | – | 81% | 63% | 52% | 37% | 33% | 29% | 25% | 18% | 16% |
| MACER Zhai et al. (2020) | 81% | 71% | 59% | 47% | 39% | 33% | 29% | 23% | 19% | 17% |
| Consistency Jeong & Shin (2020) | 78 % | 69% | 58% | 49% | 38% | 34% | 30% | 25% | 20% | 17% |
| SmooothMix Jeong et al. (2021) | 77% | 68% | 58% | 48% | 37% | 32% | 26% | 20% | 17% | 15% |
| Boosting Horváth et al. (2021) | 83% | 71% | 60% | 52% | 39% | 34% | 30% | 25% | 20% | 17% |
| DRT Yang et al. (2021) | 73 % | 67% | 60% | 51% | 40% | 36% | 30% | 24% | 20% | – |
| Black-box Zhang et al. (2020) | – | 61% | 46% | 37% | 25% | 19% | 16% | 14% | 11% | 9% |
| Data-depend Alfarra et al. (2020) | 82% | 68% | 53% | 44% | 32% | 21% | 14% | 8% | 4% | 1% |
| Sample-wise Wang et al. (2020) | 84% | 74% | 61% | 52% | 45% | 41% | 36% | 32% | 27% | 23% |
| Input-depend Súkeník et al. (2021) | 83% | 62% | 43% | 27% | 18% | 11% | 5% | 2% | 0% | 0% |
| Denoise 1 Carlini et al. (2023) | 80% | 70% | 55% | 48% | 37% | 32% | 29% | 25% | 15% | 14% |
| Denoise 2 Zhang et al. (2023) | 85% | 76% | 66% | 57% | 44% | 37% | 31% | 25% | 22% | 20% |
| ANCER Eiras et al. (2022) | 86% | 85% | 77% | – | 53% | – | 31% | – | 17% | – |
| Ours (certified accuracy w.r.t. radius) | 85% | 83% | 81% | 80% | 77% | 75% | 73% | 70% | 68% | 65% |
| Ours (certified accuracy w.r.t. ALM) | 86% | 84% | 82% | 80% | 74% | 71% | 68% | 65% | 61% | 57% |
| Improvement over Baseline (%) | +0% | -1.2% | +6.5% | +40.4% | +45.3% | +82.9% | +102.8% | +118.8% | +151.9% | +182.6% |

Table 4: Certified accuracy vs. $\ell_2$ perturbations (ImageNet).

| Radius and ALM (equivalent for isotropic) | 0.00 | 0.50 | 1.00 | 1.50 | 2.00 | 2.50 | 3.00 | 3.50 |
|---|---|---|---|---|---|---|---|---|
| Cohen's Cohen et al. (2019) | 67% | 49% | 37% | 28% | 19% | 15% | 12% | 9% |
| SmoothAdv Salman et al. (2019) | 67% | 56% | 45% | 38% | 28% | 26% | 20% | 17% |
| MACER Zhai et al. (2020) | 68% | 57% | 43% | 37% | 27% | 25% | 20% | – |
| Consistency Jeong & Shin (2020) | 57% | 50% | 44% | 34% | 24% | 21% | 17% | – |
| SmoothMix Jeong et al. (2021) | 55% | 50% | 43% | 38% | 26% | 24% | 20% | – |
| Boosting Horváth et al. (2021) | 68% | 57% | 45% | 38% | 29% | 25% | 21% | 19% |
| DRTYang et al. (2021) | 50% | 47% | 44% | 39% | 30% | 29% | 23% | – |
| Black-box Zhang et al. (2020) | – | 50% | 39% | 31% | 21% | 17% | 13% | 10% |
| Data-depend Alfarra et al. (2020) | 62% | 59% | 48% | 43% | 31% | 25% | 22% | 19% |
| Denoise 1 Carlini et al. (2023) | 48% | 41% | 30% | 24% | 19% | 16% | 13% | – |
| Denoise 2 Zhang et al. (2023) | 66% | 59% | 48% | 40% | 31% | 25% | 22% | – |
| ANCER Eiras et al. (2022) | 70% | 70% | 62% | 61% | 42% | 36% | 29% | – |
| Ours (certified accuracy w.r.t. radius) | 65% | 62% | 58% | 53% | 50% | 46% | 43% | 38% |
| Ours (certified accuracy w.r.t. ALM) | 71% | 66% | 62% | 58% | 54% | 51% | 47% | 42% |
| Improvement over Baseline (%) | +1.4% | -5.7% | +0% | -4.9% | +28.6% | +41.7% | +62.1% | +121.1% |

On all the three datasets, UCAN significantly boosts the certified accuracy. For instance, it achieves the improvement of $142.5\%$, $182.6\%$, and $121.1\%$ over the best baseline on MNIST, CIFAR10, and ImageNet, respectively. UCAN also achieves the best trade-off between certified accuracy and ALM/radius (*two important metrics*): 1) UCAN presents both larger radius/ALM and higher certified accuracy in general, and 2) On large ALM/radius, UCAN can still achieve high certified accuracy.

Finally, some examples of anisotropic vs. isotropic noise and the results for efficiency are given in Appendix G and H.

## 6 CONCLUSION

In this paper, we propose a novel randomized smoothing framework called UCAN. UCAN can transform any randomized smoothing scheme with isotropic noise into randomized smoothing with anisotropic noise with robustness guarantees. Extensive experimental results validate that UCAN significantly boosts the certified robustness of existing randomized smoothing methods.

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

## A    PROOF OF THEOREM 3.2

We restate Theorem 3.2:

**Theorem 3.2** (**Anisotropic Randomized Smoothing via Universal Transformation**). *Let $f : \mathbb{R}^d \to \mathcal{C}$ be any deterministic or random function. Suppose that for the multivariate random variable with isotropic noise $X = x + \epsilon$ in Theorem 3.1, the certified radius function is $R(\cdot)$. Then, for the corresponding anisotropic input $Y = x + \epsilon^\top \sigma + \mu$, if there exist $c'_A \in \mathcal{C}$ and $\underline{p_A}', \overline{p_B}' \in [0, 1]$ such that:*

$$\mathbb{P}(f(Y) = c'_A) \geq \underline{p_A}' \geq \overline{p_B}' \geq \max_{c \neq c'_A} \mathbb{P}(f(Y) = c) \tag{3}$$

*then $g'(x + \delta') \equiv \arg\max_{c \in \mathcal{C}} \mathbb{P}(f(Y + \delta') = c) = c'_A$ for all $||\frac{\delta'_i}{\sigma_i}||_p \equiv (\sum_i^d (\frac{\delta'_i}{\sigma_i})^p)^{\frac{1}{p}} \leq R(\underline{p_A}', \overline{p_B}')$ where $g'$ denotes the smoothed classifier based on anisotropic noise, $\delta'$ the perturbation on $x$, and $i$ the dimension index.*

*Proof.* Let $X = x + \epsilon, X \in \mathbb{R}^d$ be a multivariate random variable where $\epsilon$ follows isotropic noise distribution. Let $Y = x + \epsilon^\top \sigma + \mu, Y \in \mathbb{R}^d$ be a multivariate random variable following anisotropic noise distribution. Given the input $x$, anisotropic multiplier $\sigma$, and the mean offsets $\mu$, let $\mu' = \mu + x - x^\top \sigma$, so the anisotropic input can be written as $Y = (x + \epsilon)^\top \sigma + \mu' = X^\top \sigma + \mu'$

Define a transformation $h$ as:

$$h(z) = z^\top \sigma + \mu' \tag{6}$$

Given any deterministic or random function $f : \mathbb{R}^d \to \mathcal{C}$, define the anisotropic smoothed classifier as:

$$g'(x) = \arg\max_{c \in \mathcal{C}} \mathbb{P}(f(x + \epsilon^\top \sigma + \mu) = c) \tag{7}$$

Suppose that for an anisotropic random variable $Y$, there exist $c'_A \in \mathcal{C}$ and $\underline{p_A}', \overline{p_B}' \in [0, 1]$ such that:

$$\mathbb{P}(f(Y) = c'_A) \geq \underline{p_A}' \geq \overline{p_B}' \geq \max_{c \neq c'_A} \mathbb{P}(f(Y) = c) \tag{8}$$

By transformation $h$, the satisfied condition Eq. (8) is equivalent to:

$$\mathbb{P}(f(Y) = c'_A) = \mathbb{P}(f(X^\top \sigma + \mu') = c'_A) \tag{9}$$
$$= \mathbb{P}(f(h(X)) = c'_A) \tag{10}$$
$$\geq \underline{p_A}' \geq \overline{p_B}' \geq \max_{c \neq c'_A} \mathbb{P}(f(h(X)) = c) \tag{11}$$

Consider a new classifier $f'(x) = f(h(x))$ that maps any isotropic input to class space $\mathcal{C}$, Eq. (10) and (11) can be written as:

$$\mathbb{P}(f'(X) = c'_A) \geq \underline{p_A}' \geq \overline{p_B}' \geq \max_{c \neq c'_A} \mathbb{P}(f'(X) = c) \tag{12}$$

This is the prerequisite condition in any isotropic randomized smoothing theory as defined in Definition 3.1, so we can obtain the guarantee with isotropic certified radius $R$ such that:

$$\arg\max_{c \in \mathcal{C}} \mathbb{P}(f'(X + \delta) = c) = c'_A \tag{13}$$

$$s.t. \ ||\delta||_p \leq R(\underline{p_A}', \overline{p_B}') \tag{14}$$

By Eq. (6) and $Y = X^\top \sigma + \mu'$, the Eq. (13) is equivalent to:

$$\arg\max_{c \in \mathcal{C}} \mathbb{P}(f'(X + \delta) = c) = \arg\max_{c \in \mathcal{C}} \mathbb{P}(f(h(X + \delta)) = c) \tag{15}$$

$$= \arg\max_{c \in \mathcal{C}} \mathbb{P}(f((X + \delta)^\top \sigma + \mu') = c) \tag{16}$$

$$= \arg\max_{c \in \mathcal{C}} \mathbb{P}(f(Y + \delta^\top \sigma)) = c) = c'_A \tag{17}$$

Define $\delta' = \delta^\top \sigma$, with Eq. (17), the guarantee in Eq. (13) and (14) is equivalent to:

$$\arg\max_{c \in \mathcal{C}} \mathbb{P}(f(Y + \delta') = c) = c'_A \tag{18}$$

$$s.t. \ ||\frac{\delta'_i}{\sigma_i}||_p \leq R(\underline{p_A}', \overline{p_B}') \tag{19}$$

By Eq. (7) we have $g'(x + \delta') = \arg\max_{c \in \mathcal{C}} \mathbb{P}(f(Y + \delta') = c) = c'_A$ for all $||\frac{\delta'_i}{\sigma_i}||_p \equiv (\sum_i^d (\frac{\delta'_i}{\sigma_i})^p)^{\frac{1}{p}} \leq R(\underline{p_A}', \overline{p_B}')$.

This completes the proof.

$\square$

## B  BINARY CASE OF THEOREM 3.2

**Theorem B.1 (Universal Transformation for Anisotropic Noise).** *Let $f : \mathbb{R}^d \to \mathcal{C}$ be any deterministic or random function. Suppose that for the isotropic input $X$ in Definition 3.1, the certified radius function for the binary case is $R(\cdot)$. Then, for the corresponding anisotropic input $Y$ where $Y_i = x_i + \epsilon_i \sigma_i + \mu_i$, if there exist $c'_A \in \mathcal{C}$ and $\underline{p_A}' \in (1/2, 1]$ such that:*

$$\mathbb{P}(f(Y) = c'_A) \geq \underline{p_A}' \geq \frac{1}{2} \tag{20}$$

*Then $g'(x + \delta') = \arg\max_{c \in \mathcal{C}} \mathbb{P}(f(Y) = c) = c'_A$ for all $||\delta'_i/\sigma_i||_p \leq R(\underline{p_A}')$ where $g'$ denotes the anisotropic smoothed classifier, $\delta'$ denotes the perturbation injected to $g'$.*

*Proof.* The proof for the binary case is similar to the proofs for the multiclass-case (See Appendix A).  $\square$

## C  NEW ACCURATE METRIC FOR CERTIFIED REGION

How to develop a general metric for evaluating the robustness region for randomized smoothing with anisotropic noise is an important but challenging problem. We observe that the guarantee in Theorem 3.2 forms a certified region, within which the perturbation is certifiably safe to the smoothed classifier. The $\ell_p$-norm bounding on the scaled perturbation $\delta'_i/\sigma_i$ results in the anisotropy of the certified region around the input. We illustrate the anisotropic certified region for different $\ell_p$-norm guarantees in Figure 1(b). It shows that if the $\delta$ space is a 2-dimension space, then the guarantee of anisotropic RS draws a rhombus, ellipse, and rectangle in $\ell_1$, $\ell_2$, and $\ell_\infty$ norms, respectively. Within the anisotropic region, we can find an isotropic region that also satisfies the robustness guarantee (a subset of the anisotropic region), which results in an explicit certified radius (see Corollary 3.3).

However, while evaluating the performance of randomized smoothing with anisotropic noise via Eq. (4) in Corollary 3.3, although is correct and explicit, is inaccurate since it only captures a subset of the certified region and ignores the anisotropy of the guarantee (such radius is *sufficient but not necessary* to the certified guarantee).

Specifically, Eq. (4) evaluates the performance only based on the blue region in Figure 1(b), but the guarantee in Theorem 3.2 actually guarantees that all the $\delta$ (perturbations) within the green region are safe. Therefore, to fairly and accurately evaluate the performance of certification via anisotropic noise, we need to develop a novel metric that can cover the entire certified region in highly-dimensional $\delta$ space, and also cover the adversarial perturbations in all $\ell_p$-norms.

From another perspective, evaluating the performance of randomized smoothing can be considered as evaluating the size of the robust perturbation set $S(n, p)$.

**Definition C.1.** The $d$-dimensional robust perturbation set is defined as

$$S(d, p) \equiv \{(\delta_1, \delta_2, ..., \delta_d) : ||\frac{\delta_i}{\sigma_i}||_p \leq R, \ p > 0\} \tag{21}$$

Consider the Euclidean structure, $S(d, p)$ is a finite set in $d$-dimensional Euclidean space. Therefore, we leverage the Lebesgue measure Bartle (2014) to compute the size of $S(d, p)$ (see Theorem C.2).

**Theorem C.2** (**Lebesgue Measure of the Robust Perturbation Set** $S(d, p)$)**.** *Let $S(d, p)$ be defined as in Eq. (21), then the Lebesgue measure of the robust perturbation set is given by*

$$V_S(d, p) = \frac{(2R\Gamma(1 + \frac{1}{p}))^d \prod_{i=1}^{d} \sigma_i}{\Gamma(1 + \frac{d}{p})} \tag{22}$$

*where $\Gamma$ is the Euler gamma function defined in Definition D.1.*

*Proof.* See the detail proof in Appendix D                                    □

We observe that for a fixed $d$ and $p$, the $\frac{(2\Gamma(1+\frac{1}{p}))^d}{\Gamma(1+\frac{d}{p})}$ factor in the Lebesgue measure is a constant. Then, when comparing the Lebesgue measure in the same norm $\ell_p$ and the same space $\mathbb{R}^d$, the constant term can be ignored. Also, the $R^d$ factor can lead to infinite numeral computation, thus we also scale the Lebesgue measure by calculating the $d$-th root. As a result, we define the Alternative Lebesgue Measure (ALM) of the robust perturbation set with the same $d$ and $p$ as:

$$V_S' = \sqrt[d]{\prod_{i=1}^{d} \sigma_i} R \tag{23}$$

**Alternative Lebesgue Measure**[2] **vs. Isotropic Radius**. When the multipliers of the scale parameter for anisotropic noise $\sigma_1 = \sigma_2 = ... = 1$, the noise turns into the isotropic noise and the alternative Lebesgue measure turns into the certified radius $R$. Therefore, the alternative Lebesgue measure can be treated as a generalized metric compared to the certified radius. This generalization based on the certified radius also enables us to fairly compare the randomized smoothing based on anisotropic noise with isotropic noise.

Note that the new metric ALM is not developed to bound the perturbation, but to accurately measure the certified guarantees of randomized smoothing with anisotropic noise.

## D    PROOF OF THEOREM C.2

*Proof.*

---

[2]ALM is like the normalized radius of the certified region in all $d$ dimensions.

**Definition D.1 (Euler Gamma Function).** The Euler gamma function is defined by

$$\Gamma(\beta) = \int_0^\infty \alpha^{\beta-1} e^{-\alpha} d\alpha \tag{24}$$

There are some properties of $\Gamma$: 1) For all $\beta > 0$, $\beta\Gamma(\beta) = \Gamma(\beta + 1)$, 2) For all positive integers $n$, $\Gamma(n) = (n-1)!$, and 3) $\Gamma(1/2) = \sqrt{\pi}$.

**Definition D.2 (n-dimensional Generalized Super-ellipsoid).** The d-dimensional generalized super-ellipsoid ball is defined as

$$E(d, p) = \{(\delta_1, \delta_2, ..., \delta_d) : \sum_{i=1}^{d} |\frac{\delta_i}{c_i}|^{p_i} \leq 1, p_i > 0\} \tag{25}$$

**Lemma D.3 (Lebesgue Measure of Generalized Super-ellipsoids Ahmed & Saleeby (2018)).** *The Lebesgue measure of the generalized super-ellipsoids is given by*

$$V_E(d, p) = 2^d \frac{\prod_{i=1}^{d} c_i \Gamma(1 + \frac{1}{p_i})}{\Gamma(1 + \sum_{i=1}^{d} \frac{1}{p_i})}; p_i > 0, d = 1, 2, 3, ... \tag{26}$$

We leverage Lemma D.3 to prove Theorem 3.2. Let $p_i = p$, the d-dimensional robust perturbation set is equivalent to

$$S(d, p) = \{(\delta_1, \delta_2, ..., \delta_d) : \sum_{i=1}^{d} |\frac{\delta_i}{\sigma_i R}|^p \leq 1\} \tag{27}$$

The Lebesgue measure in Lemma D.3 will be

$$V_E(d, p) = 2^d \frac{\prod_{i=1}^{d} c_i R \Gamma(1 + \frac{1}{p})}{\Gamma(1 + \sum_{i=1}^{d} \frac{1}{p})} = \frac{(2R\Gamma(1 + \frac{1}{p}))^d \prod_{i=1}^{d} \sigma_i}{\Gamma(1 + \frac{d}{p})};$$
$$p > 0, d = 1, 2, 3, ... \tag{28}$$

Thus, this completes the proof. $\square$

## E    Input-dependent Noise Parameter Generator (NPG)

It consists of 4 convolutional layers followed by leaky-ReLU Xu et al. (2015). Similar to the generator in universal anisotropic noise, the output is rectified by the amplified hyperbolic tangent function to stabilize the training process. Note that our parameter generator is a small network (5 layers), thus it can be plugged in before any classifier for generating the input-dependent anisotropic noise without consuming too many computing resources (see Section H for a detailed discussion on running time). For both $\mu$ and $\sigma$, we train different parameter generators using the same architecture.

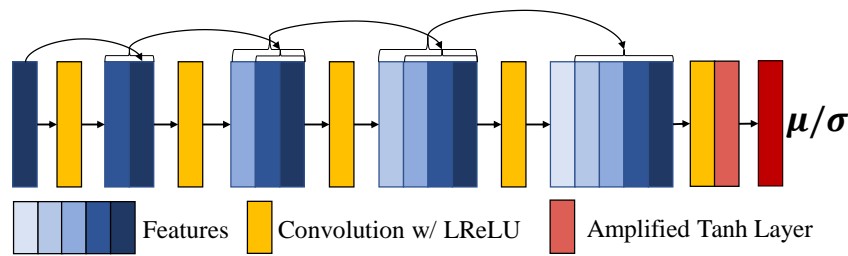

Figure 6: **Architecture of parameter generator for input-dependent anisotropic noise**

# F   PRACTICAL ALGORITHMS

Following Cohen et al. Cohen et al. (2019), we also use the Monte Carlo algorithm to bound the prediction probabilities of smoothed classifier and compute the ALM (certified region). Different from Cohen et al. Cohen et al. (2019), our noise distributions are either pre-assigned (as pattern-fixed) or produced by the parameter generator (either universal or input-dependent). Our algorithms for certification and prediction using different noise generation methods are summarized in Algorithm 1 and 2 (w.l.o.g., taking the binary classifier as an example).

For simplicity of notations, the generation of anisotropic $\mu$ and $\sigma$ are summarized by the noise generation method(s) $M$. In case of pattern-fixed anisotropic noise, $M$ outputs pre-assigned fixed variance and zero-means; in case of universal and input-dependent anisotropic noise, $M$ adopts the parameter generators to generate the mean and variance maps. In the certification (Algorithm 1), we select the top-1 class $\hat{c_A}$ by the CLASSIFYSAMPLES function, in which the base classifier outputs the prediction on the noisy input sampled from the noise distribution. Once the top-1 class is determined, classification will be executed on more samples and the LOWERCONFBOUND function will output the lower bound of the probability $\underline{p_A}'$ computed by the Binomial test. If $\underline{p_A}' > \frac{1}{2}$, we output the prediction class and the ALM (measuring the certified region). Otherwise, it outputs ABSTAIN. In the prediction (Algorithm 2), we also generate the noise and then compute the prediction counts over the noisy inputs. If the Binomial test succeeds, then it outputs the prediction class. Otherwise, it returns ABSTAIN.

---

**Algorithm 1** UCAN-Certification

---

    **Given:** Base classifier $f$, anisotropic noise generation method $M$, input (e.g., image) $x$, number of Monte Carlo samples $n_0$ and $n$, confidence $1 - \alpha$

1: $\mu, \sigma \leftarrow M$
2: $counts\_select \leftarrow$ CLASSIFYSAMPLES$(f, x, \mu, \sigma, n_0)$
3: $\hat{c}_A \leftarrow$ **top index in** $counts\_select$
4: $counts \leftarrow$ CLASSIFYSAMPLES$(f, x, \mu, \sigma, n)$
5: $\underline{p_A}' \leftarrow$ LOWERCONFBOUND$(counts[\hat{c}_A], n, 1 - \alpha)$
6: **if** $\underline{p_A}' > \frac{1}{2}$ **then**
7:     **return** prediction $\hat{c}_A$ and ALM (certified region) $\sqrt[d]{\prod_i \sigma_i} R$
8: **else**
9:     **return** ABSTAIN
10: **end if**

---

**Algorithm 2** UCAN-Prediction

---

    **Given**: Base classifier $f$, anisotropic noise generation method $M$, input (e.g., image) $x$, number of Monte Carlo samples $n$, confidence $1 - \alpha$

1: $\mu, \sigma \leftarrow M$
2: $counts \leftarrow$ CLASSIFYSAMPLES$(f, x, \mu, \sigma, n)$
3: $\hat{c}_A \leftarrow$ **top index in** $counts$
4: $n_A \leftarrow counts[\hat{c}_A]$
5: **if** BINOMIALPVALUE$(n_A, n, 0.5) \leq \alpha$ **then**
6:     **return** prediction $\hat{c}_A$
7: **else**
8:     **return** ABSTAIN
9: **end if**

---

# G   VISUALIZATION

We present several examples of anisotropic and isotropic noise in Figure 7. All the proposed anisotropic noise generation methods find better spatial distribution to generate the anisotropic noise. Both the pattern-fixed and universal anisotropic reduce the variance on the key area, except the universal anisotropic noise on ImageNet (it seems the parameter generator does not find a constant key area on ImageNet due to the complicated data distribution in ImageNet). We also observe that the parameter generator for input-dependent anisotropic noise generates large mean offsets to compensate

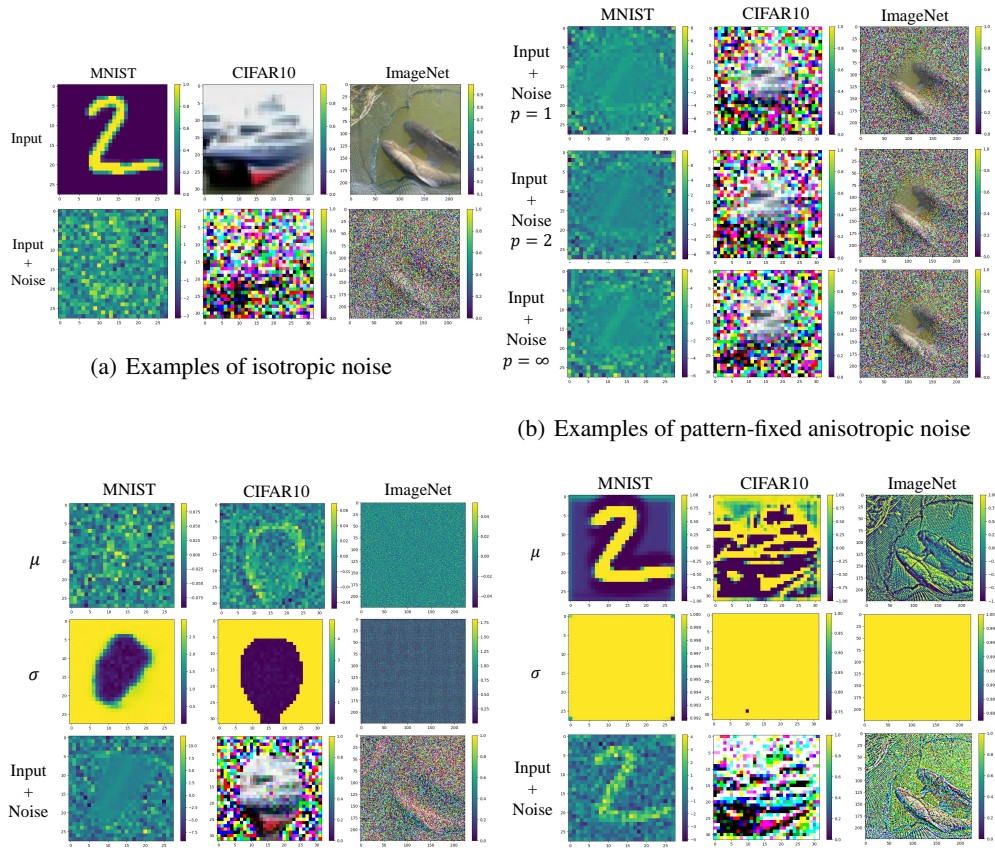

Figure 7: Visualized examples of anisotropic noise vs. isotropic noise. The visualization are based on the input examples in (a).

for the high $\sigma$ values. It turns out that with high variance ($\sigma_i \approx 1$), the object is still recognizable in the input-dependent anisotropic noise.

## H  EFFICIENCY

UCAN is a universal framework that can be readily integrated into existing randomized smoothing to boost performance. Whether the extra neural network components (parameter generator) in UCAN will degrade the efficiency of existing randomized smoothing is an important question. We show that the running time overhead resulting from the parameter generator is negligible compared to the running time of the certification, since for each input, the classifier needs to evaluate $N$ noise samples while $\mu$ and $\sigma$ are generated once. Typically, $N = 100,000$. UCAN can be trained offline and tested online to boost the performance of randomized smoothing. We evaluate the online certification running time for input-dependent anisotropic noise generation and traditional randomized smoothing Cohen et al. (2019) on ImageNet with four Tesla V100 GPUs and $2,000$ batch size, the average runtimes over 500 samples are 27.43s and 27.09s per sample for our method and Cohen et al. (2019)'s method, respectively. Thus, the NPG will only slightly increase the overall runtime.

## I  EXPERIMENT DETAILS

We use the original size of the images in MNIST and CIFAR10, i.e., $28 \times 28$ and $3 \times 32 \times 32$, respectively. For the ImageNet dataset, we resize the images to $3 \times 224 \times 224$. In the training, we

train the base classifier and the parameter generator (if needed) with all the training set in three datasets. For the MNIST dataset, we use a simple two-layer CNN as the base classifier. For the CIFAR10 and ImageNet datasets, we use the ResNet110 and ResNet50 He et al. (2016) as the base classifier, respectively.

