# OpenReview forum: "Universally Amplifying Randomized Smoothing for Certified Robustness with Anisotropic Noise"
_ICLR.cc/2024/Conference — Submitted to ICLR 2024_

### Official Review · Reviewer_SMHf · 2023-10-19

**Soundness:** 1 poor
**Presentation:** 3 good
**Contribution:** 2 fair
**Rating:** 3
**Confidence:** 3

**Summary:**

This paper proposes an approach to Certified Robustness via Randomized Smoothing, in which the noise distribution for the different data dimensions is allowed to vary, i.e. is anisotropic. Three example noise parameter generators are proposed, and experimental results given on standard datasets against SOTA methods.

**Strengths:**

The paper is well presented, clearly structured and well written, with a reasonably comprehensive set of experiments performed.

**Weaknesses:**

My scoring reflects very much an issue raised in the Questions below. Clarification on that from the authors could substantially change my assessment.

The paper presents only an incremental improvement on SOTA methods, and in many ways the main Theorem (3.2) is really no more than a simple Corollary from prior work with a simple affine transformation on the variables (which would usually be done anyway in dealing with non-image data with very different means and scales in each data dimension).

The main issues, written in questions to the authors, concern (a) the presentation of the comparison with isotropic SOTA, and (b) the validity of the input-dependent noise (the method introduced in section 4.3). This latter is also used as the basis for the main results in section 5.3. Hence my rating for the paper could be improved significantly in light of any author response/clarification to my questions.

Some more minor comments here though:
- The binary case rather than multi-class is used for Cohen. This gives poorer results, of course. Not an issue, as it is done consistently for the proposed and SOTA methods. But it should be at least clarified as the statement of Theorem 3.2 suggests that the p_B value will be used, but then in Table 1 it is not (ie it is replaced by p_B = 1 – p_A)
- The definition of Acc in the Metrics section has it defined as a function of V’_S. And yet V’_S does not appear on the RHS!! (instead you have replaced it with the dth root of the product of the sigmas times R)
- At small radii, the SOTA methods are better (on the graphs) than the proposed method. Some discussion would be welcome on this.
- In section 5.1, it is mentioned that Cohen gives a tight radius. Again, as per above, this is really only in the non-binary form.

**Questions:**

In the experiments, it seems that the results vs {min \sigma_i} R should be presented (per Corollary 3.3) for a fair comparison, i.e. it is not clear to me that the proposed technique certifies a strictly larger L_p ball in the same conditions than SOTA (and what “same conditions” may mean is not clear e.g. it may mean isotropic sigma=1 and \product \sigma_i = 1 in anisotropic case). That said, the results in section 5.3 apparently include a certified accuracy wrt radius. It is not clear if this may be an answer to my query as it is not clear exactly what is being reported here.

Re section 4.3, I am a bit confused. This input-dependent proposal would give an x-dependent sigma, and hence an x-dependent classifier (ie x-dependent noise, over and above the obvious dependence on x). Theorem 3.2 holds assuming the noise is constant in the ball around x. This proposal in 4.3 violates that surely. Hence it is no longer true that Theorem 3.2 guarantees that the classifier gives an unchanged output in the region claimed, and so the proposed classifier is not certified robust in the claimed reghion around x. I may have misunderstood. Clarification/explanation is welcome.
Please see aeXiv paper 2110.05365 for an example of work done with input-dependent noise.

---

> ### Author Response · Authors · 2023-11-15
>
> Thank you for the insightful comments and please find our clarifications as below. We also make more clarifications in the paper to avoid confusion/misunderstanding.
>
> 1. **On the Novelty of Theorem 3.2**:
>
> > The paper presents only an incremental improvement on SOTA methods, and in many ways the main Theorem (3.2) is really no more than a simple Corollary from prior work with a simple affine transformation on the variables (which would usually be done anyway in dealing with non-image data with very different means and scales in each data dimension).
>
> We respectfully clarify that the derived universal theories are not simple (by covering both strict robustness w.r.t. heterogeneous dimensions and universality), as detailed in Appendix A. Even if its conclusion might be considered as simple, as noted by Reviewer v1kJ, simple-but-effective solutions are not generally considered as ``weakness'', but would be desirable in practice.
>
> 2. **Clarification on Isotropic Results**:
>
> > In the experiments, it seems that the results vs $\{min \sigma_i\} R$  should be presented (per Corollary 3.3) for a fair comparison, i.e. it is not clear to me that the proposed technique certifies a strictly larger $L_p$ ball in the same conditions than SOTA (and what “same conditions” may mean is not clear e.g. it may mean isotropic $\sigma=1$ and $\prod \sigma_i = 1$ in anisotropic case). That said, the results in section 5.3 apparently include a certified accuracy wrt radius. It is not clear if this may be an answer to my query as it is not clear exactly what is being reported here.
>
> In our experimental results, the “certified accuracy w.r.t radius” refers to certified accuracy w.r.t. $\min\{\sigma_i\}R$, which equals to the $\ell_p$ radius (e.g., the Gaussian $\ell_2$ case). Therefore, we did achieve a strictly larger $\ell_2$-ball than other SOTAs. In the entire paper (including Section 5.3), all the “radius” means traditional radius for the $\ell_p$ ball.
>
> 3. **Input-dependent Certification Validity**:
>
> > Re section 4.3, I am a bit confused. This input-dependent proposal would give an x-dependent sigma, and hence an x-dependent classifier (ie x-dependent noise, over and above the obvious dependence on x). Theorem 3.2 holds assuming the noise is constant in the ball around x. This proposal in 4.3 violates that surely. Hence it is no longer true that Theorem 3.2 guarantees that the classifier gives an unchanged output in the region claimed, and so the proposed classifier is not certified robust in the claimed region around x.
>
> Regarding the input-dependent certification, we noticed the discussion in Eiras et al. (2021). We respectfully clarify that our input-dependent noise method is still sound and does not breach certification principles.
>
> The $x$-dependent classifier is fixed in the certification on $x$, that means with $g(x,\mu(x),\sigma(x))$ and $R$, we are guaranteeing $g(x+\delta,\mu(x),\sigma(x))=g(x,\mu(x),\sigma(x))$ as described in Theorem 3.2 (note that noise is pre-computed and $\mu(x),\sigma(x)$ are actually constants). We do not change the $x$-dependent classifier during the certification, thus we achieve consistent prediction over such a specific classifier. This does not violate the certification of randomized smoothing, instead, it sticks to the inherent characteristics of the RS-based certification, i.e., input-dependence (The RS certification is inherently input-dependent since the radius $R(x)$ only works on the specific input $x$ and a specific classifier). Please also see the detailed explanation in our ``Common Concerns'' response.
>
> 4. **Responses to Minor Comments**:
>
> - Binary Case: We appreciate this observation and have revised the caption of Table 1 accordingly.
>
> - Metrics Clarification: The metric named $V'_S$ is referred to as ALM throughout the paper. We have amended the symbols to ALM for consistency and clarity.
>
> - Performance at Small Radii: The reason SOTA methods perform better at smaller radii is due to the noise variance-prediction accuracy trade-off. Our method, with single $\lambda\approx 1$, competes with SOTA methods at various $\lambda$ settings. Notably, our Input-dependent method outperforms these trade-offs even with a single setting.
>
> - Cohen's Tight Radius: We have revised the statement about Cohen's method to reflect its application more accurately.

---

> > ### Comment · Reviewer_SMHf · 2023-11-16
> >
> > Thank you for your responses. In line with reviewer v1kj I still am not convinced of the validity of the certification in the input-dependent case and maintain my score.

---

### Official Review · Reviewer_v1kJ · 2023-10-27

**Soundness:** 1 poor
**Presentation:** 3 good
**Contribution:** 2 fair
**Rating:** 3
**Confidence:** 4

**Summary:**

This paper proposes to shift and re-scale the noise in randomized smoothing in order to generate anisotropic robustness guarantees. The approach is universal in the sense that any randomized smoothing-based method can be transformed into a model with such anisotropic guarantees. Experiments on benchmark image classification datasets demonstrate increased certified accuracy curves compared to past works.

**Strengths:**

1. The paper is very easy to read.
2. The approach is simple to understand, and the resulting theoretical guarantee follows from past robustness guarantees in a very straightforward manner. This simplicity should be considered a strength of the method, not a weakness of the paper.
3. The experiments are thorough, with comparisons to a wide range of prior methods and on high-dimensional image datasets (e.g., ImageNet).

**Weaknesses:**

See "Questions" section below.

**Questions:**

1. "However, its theory is based on assumptions and the universality is relatively limited." What assumptions? Please at least briefly mention them and why they are stringent.
2. Definition 3.1 does not really appear to be a "definition" in mathematical terms. It looks more like you are re-stating the certified radius theorems for general distributions and norms. So, this should probably be labeled as a "theorem".
3. Please move the definition/review of alternative Lebesgue measure to Section 3.1, where Table 1 appears with Alt. Lebesgue Measure as a column.
4. In Section 4.2, the loss function is solely a function of the NPG parameters $\theta_g$, correct? If so, it would be good to explicitly write $\mathcal{L}(\theta_g)$ to emphasize to the reader what you are optimizing over. Furthermore, $\sigma$ and $\mu$ would be functions of this parameter $\theta_g$, right? If so, it would also be good to write $\sigma(\theta_g)$ and $\mu(\theta_g)$ in the smoothing loss expression.
5. MOST IMPORTANT PROBLEM: Your highest performing approach, using input-dependent anisotropic smoothing parameters that are optimized per-input, breaks the robustness certificates. Namely, randomized smoothing robustness certificates intimately rely on the same model being used to predict at the nominal point $x$ and all perturbed versions $x+\delta$ in the certified ball around $x$. However, if you optimize $\mu,\sigma$ at $x$, then the smoothing-based certificate only says that $x+\delta$ will yield the same prediction if you also use the same parameters $\mu,\sigma$ to define the prediction at $x+\delta$. But, according to your scheme, you actually re-optimize $\mu,\sigma$ at the perturbed test input $x+\delta$ to generate the prediction at $x+\delta$, meaning you are using a different model than what smoothing certifies at $x$. This mathematical breakdown of certified robustness for input-dependent smoothing has been noted before in past works, and is the reason why works like Eiras et al. (2022) augment their input-dependent scheme with "memory." In order for your input-dependent smoothing scheme to work, you would also need to appeal to some "fix" like this memory method, which comes with its own issues (e.g., relating to dependency on input order, and increased memory overhead costs). Either you should fix this issue (and hopefully your certificates still provide substantial improvement over state-of-the-art), or you should remove this input-dependent part of the paper (which, in my opinion, would significantly reduce the contributions of the paper).

---

> ### Author Response · Authors · 2023-11-15
>
> Thank you for the insightful comments and please find our clarifications as below. We also make more clarifications in the paper to avoid confusion/misunderstanding.
>
> 1. **On the Presenting of Related Works**:
>
> > "However, its theory is based on assumptions and the universality is relatively limited." What assumptions? Please at least briefly mention them and why they are stringent.
>
> Our approach does not need the assumptions in Eiras et al., where the classifier's $L$-Lipschitz continuity is a prerequisite. This assumption imposes limitations on the universality of their theorem. Conversely, our method is designed to work universally with any classifier, thereby offering broader applicability.
>
> 2 \& 3. **Regarding the Paper Revisions**:
>
> > Definition 3.1 does not really appear to be a "definition" in mathematical terms. It looks more like you are re-stating the certified radius theorems for general distributions and norms. So, this should probably be labeled as a "theorem".
> > Please move the definition/review of alternative Lebesgue measure to Section 3.1, where Table 1 appears with Alt. Lebesgue Measure as a column.
>
> We appreciate your suggestions for improving the paper's structure and clarity. We have revised the paper accordingly, particularly in relabeling Definition 3.1 as a theorem and relocating the alternative Lebesgue measure definition to Section 3.1, as suggested.
>
> 4. **Clarifications in Section 4.2**:
>
> > In Section 4.2, the loss function is solely a function of the NPG parameters $\theta_g$, correct? If so, it would be good to explicitly write $\mathcal{L}(\theta_g)$ to emphasize to the reader what you are optimizing over. Furthermore, $\sigma$ and $\mu$ would be functions of this parameter $\theta_g$, right? If so, it would also be good to write $\sigma(\theta_g)$ and $\mu(\theta_g)$ in the smoothing loss expression.
>
> We have updated our paper to clarify that the loss function is optimized over both $\theta_f$ and $\theta_g$. Additionally, we have made it explicit that $\mu$ and $\sigma$ are functions of the parameter $\theta_g$. These revisions should enhance the clarity of our methodology.
>
> 5. **On the Validity of Input-Dependent Randomized Smoothing**:
>
> > Your highest performing approach, using input-dependent anisotropic smoothing parameters that are optimized per-input, breaks the robustness certificates.
>
> We noticed the discussion in Eiras et al. (2022) and understand your concerns regarding our input-dependent Randomized Smoothing (RS) approach. However, we respectfully clarify the soundness of this method.
>
> > Namely, randomized smoothing robustness certificates intimately rely on the same model being used to predict at the nominal point $x$ and all perturbed versions $x+\delta$ in the certified ball around $x$.However, if you optimize $\mu,\sigma$ at $x$, then the smoothing-based certificate only says that $x+\delta$ will yield the same prediction if you also use the same parameters $\mu,\sigma$ to define the prediction at $x+\delta$. But, according to your scheme, you actually re-optimize $\mu,\sigma$ at the perturbed test input $x+\delta$ to generate the prediction at $x+\delta$, meaning you are using a different model than what smoothing certifies at $x$. This mathematical breakdown of certified robustness for input-dependent smoothing has been noted before in past works, and is the reason why works like Eiras et al. (2022) augment their input-dependent scheme with "memory."
>
> In this case, with $g(x,\mu(x),\sigma(x))$ and $R$ where $\mu(x),\sigma(x)$ are pre-computed (independent of $\delta$) as constants, we are guaranteeing $g(x+\delta,\mu(x),\sigma(x))=g(x,\mu(x),\sigma(x))$ as described in Theorem 3.2. Then, the models used for predicting $x$ and $x+\delta$ are the same in the certification, thus do not break the certification. The entire certification process has nothing to do with $g(x+\delta,\mu(x+\delta),\sigma(x+\delta))$ and the noise optimization is not based on $x+\delta$. Please also see the detailed explanation in our **Common Concerns** response.

---

> > ### Comment · Reviewer_v1kJ · 2023-11-15
> >
> > 1. Notice that Eiras does not assume that the base classifier $f$ is Lipschitz, so their method is "designed to work universally with any classifier" as well. They prove that their smoothed classifiers $g$ are Lipschitz with respect to the anisotropic $\ell_2$ and $\ell_\infty$ norms, no matter what the base classifier is. This is a classical property of randomized smoothing: convolving a general function with an adequately smooth probability density function or "kernel" gives rise to a new function that inherits smoothness from the kernel. Their theoretical robustness certificate holds very generally for Lipschitz classifiers, of which smoothed classifiers via randomized smoothing are a special case. Therefore, your assertion that their "theory is based on assumptions that the networks are L-Lipschitz continuous and thus the universality is relatively limited" is not well-justified (since it is implying that they assume the base classifiers are L-Lipschitz, which they do not), and I suggest removing it.
> >
> > 2,3,4. Thank you for the revisions.
> >
> > 5. I have read both your "Common Concerns" response, as well as your individual response to me. I am still not convinced that your robustness guarantee holds. Unless I'm misunderstanding something, then, at test time, the attacked input going to your model is $x+\delta$. Therefore, your NPG will output $\mu(x+\delta)$ and $\sigma(x+\delta)$ as the noise parameters. These will in general define a different smoothing distribution than that corresponding to the clean test input $x$, which has parameters $\mu(x)$ and $\sigma(x)$ output by the NPG. Therefore, the certification condition, as you call it, would need to be that $g(x+\delta,\mu(x+\delta),\sigma(x+\delta)) = g(x,\mu(x),\sigma(x))$, which is not what is guaranteed by the randomized smoothing framework. In other words, the actual output of your model at an attacked input $x+\delta$ is $g(x+\delta,\mu(x+\delta),\sigma(x+\delta))$, and your certificates do not guarantee that this output coincides with the output $g(x,\mu(x),\sigma(x))$ generated by the clean input $x$.
> >
> > Given that my primary concern (shared with Reviewer SMHf) still remains, I maintain my original score.

---

> > > ### Author Response · Authors · 2023-11-16
> > >
> > > Thanks for the response. Also, thanks for the clarification on Eiras et al. We will revise the description as suggested.
> > >
> > > Based on the response, we would like to clarify that there is a misunderstanding on the setting of the randomized smoothing.
> > >
> > > In general randomized smoothing, you are using the classifier's prediction probability $p_A$ on $x$ (empirical results) to guarantee **its prediction** on $x+\delta$ (theoretical results), you will not empirically testing $x+\delta$ but to theoretically guarantee this prediction. To further justify this, (1) if the prediction results on attacked/perturbed input $x+\delta$ rely on the empirical testing on the smoothed model (as mentioned in the response "at test time, the attacked input going to your model is $x+\delta$"), what is the meaning of guaranteeing the prediction of $x+\delta$? (2) there exists an unlimited set of $\delta$ within the radius for $x$, how to test all of them? We don't really infer $x+\delta$ in our algorithms/implementation, we theoretically guarantee its prediction.
> > >
> > > The reviewer seems to misunderstand that we are doing $g(x+\delta,\mu(x+\delta),\sigma(x+\delta))=g(x,\mu(x),\sigma(x))$, but actually, we are guaranteeing $g(x+\delta,\mu(x), \sigma(x))=g(x,\mu(x),\sigma(x))$, please check our Theorem 3.2 and see the implementation details in our code. In the implementation, we can consider the classifier and the noise generation to be independent. Given the input $x$, we fixed the noise for the whole certification. **The noise parameters are generated once at the beginning of each certification based on $x$, and fixed during the certification** to guarantee the consistent prediction result for all kinds of $x+\delta$ within R. The certification is not violated in this case.

---

> ### Comment · Reviewer_v1kJ · 2023-11-16
>
> Thank you for your response. I believe you are still misunderstanding my point. Let me try to make it more clear:
>
> I agree with you that, mathematically, the equality $g(x+\delta,\mu(x),\sigma(x)) = g(x,\mu(x),\sigma(x))$ holds. However, my main point is that **this equality does not reflect how your classifier makes its predictions in reality, and thus provides no meaningful robustness guarantee**.
>
> To see why this is the case, let $x$ be some fixed input. Your scheme then chooses a mean $\mu(x)$ and covariance $\sigma(x)$ based on this $x$. Then, to form the prediction for the input $x$, you compute $g(x,\mu(x),\sigma(x)) = \arg\max_{c\in\mathcal{C}} \mathbb{P}_{\epsilon \sim D} (f(x + \epsilon^\top \sigma(x) + \mu(x))=c)$, where $D$ is some isotropic "base distribution," such as an isotropic Gaussian. Now, your equality says that $g(x + \delta,\mu(x),\sigma(x)) = g(x,\mu(x),\sigma(x))$ for all perturbations $\delta$ inside of some region $\mathcal{R}$ of the input space containing $x$. **This does not imply that your classification scheme's prediction is constant over this region**.
>
> Specifically, consider a NEW input $x'$ (possibly different from $x$, and therefore I put the "prime") for which you are to make a prediction. Well, substituting $x'$ for $x$ in the above prediction rule (which we both have agreed upon) gives that the prediction is $g(x',\mu(x'),\sigma(x')) = \arg\max_{c\in\mathcal{C}} \mathbb{P}_{\epsilon \sim D} (f(x' + \epsilon^\top \sigma(x') + \mu(x'))=c)$. In other words, to form a prediction for the input $x'$, you again compute the input-dependent mean and covariance, and perform the RS-based prediction at $x'$ using its optimized $\mu(x')$ and $\sigma(x')$.
>
> Now, suppose that I reveal to you that actually, I was an adversary, and the input $x'$ that I gave to you and asked you to classify was $x'=x+\delta$ for some perturbation $\delta$ that I chose within your region $\mathcal{R}$. Then your prediction for this perturbed input was computed as $g(x',\mu(x'),\sigma(x'))=g(x+\delta,\mu(x+\delta),\sigma(x+\delta)) = \arg\max_{c\in\mathcal{C}} \mathbb{P}_{\epsilon \sim D} (f(x + \delta + \epsilon^\top \sigma(x+\delta) + \mu(x+\delta))=c)$, which is **not** guaranteed to be equal to the prediction $g(x,\mu(x),\sigma(x))$ corresponding to $x$. Therefore, the equality $g(x+\delta,\mu(x),\sigma(x)) = g(x,\mu(x),\sigma(x))$ does not say anything about the prediction/output $g(x+\delta,\mu(x+\delta),\sigma(x+\delta))$ of an attacked input $x+\delta$.

---

> > ### Author Response · Authors · 2023-11-16
> >
> > Thanks for the response.
> >
> > If we are understanding correctly, your comments are based on an assumption that the noise $\mu$ and $\sigma$ are changing according to the adversarial input during the certification. This is not the setting in our original submission, and we want to clarify more on this point.
> >
> > Given the input $x$ that we want to certify, we compute the noise parameter $\mu_0=\mu(x)$ and $\sigma_0=\sigma(x)$ as constants and then construct the smoothed classifier with these fixed noises. During the certification on $x$, no matter what $x+\delta$ is, the smoothed classifier is fixed on $\mu_0$ and $\sigma_0$, until the input that we want to certify changes. In other words, you can consider that we are constructing different smoothed classifiers for certifying different input $x$.
> >
> > Let's further take your example to illustrate how this can be achieved. Suppose you were the adversary, and we were the model owner.  You gave me $x'$ and asked me to classify $x'=x+\delta$ for some perturbation $\delta$ within $R$. There are two situations here:
> >
> > 1) suppose we both know $x$ and the corresponding $R$, that means we are based on the knowledge of previous certification to classify the $x'$, then we can definitely compute $\mu(x)$ and $\sigma(x)$ (since we know $x$) and predict $g(x',\mu(x),\sigma(x))$ or simply compare $||\delta||_p$ and $R$.
> >
> > 2) suppose we both don't know $x$ and $R$, then both Cohen et al. and our method cannot provide any guarantee on this prediction since the guarantee of the prediction on $x'$ is based on the certification on $x$. Without knowing $x$, the radius $R$ cannot be computed.
> >
> > We believe both Cohen et al. and our method are in situation 1). Thanks for the response and we are open to and welcome any further questions related to these points.

---

> ### Comment · Reviewer_v1kJ · 2023-11-16
>
> Thanks again for the prompt reply. First of all, with respect to your two proposed situations:
>
> Assuming that you always know the clean input $x$ is an unrealistic and very stringent assumption. If I gave you $x'=x+\delta$, how would you know what the clean input $x$ is? If this were the case, then all of the recent work on certifiably robust machine learning in the presence of $\ell_p$-norm bounded additive adversaries would be pointless, since one could simply reconstruct $x$ from $x'$, and then classify the clean input $x$ (which is much less likely to induce an error). So, this first situation is meaningless to consider in your setting of adversarial threats. Thus, the standard setting for papers in this area (and randomized smoothing, in particular), is to assume that all that you have at your disposal is whatever input you are given ($x'$ in the case of an attacked input), and you cannot distinguish whether this was attacked or not. In this standard setting, there is no possible way for you to take the attacked $x'$ as an input, and compute $\mu(x),\sigma(x)$ for the clean input.
>
> So, there is an essential question to consider that points to the flaw at hand:
> What does your "certificate" $g(x+\delta,\mu(x),\sigma(x)) = g(x,\mu(x),\sigma(x))$ represent? To me, this says that, if you predict the output of $x+\delta$ using the smoothing distribution for $x$, then the prediction will match that of $x$. The problem is twofold: 1) you cannot compute $\mu(x),\sigma(x)$ (or even samples drawn from this distribution) unless you have access to the clean input $x$, which you do not, and 2) $g(x+\delta,\mu(x),\sigma(x))$ is not even the prediction that your scheme would output if $x+\delta$ was treated as its own input, since the smoothing distribution would be re-optimized for the input $x+\delta$.

---

> > ### Author Response · Authors · 2023-11-16
> >
> > Thanks for the interesting discussion.
> >
> > It seems there is confusion between the setting of certified defense and the empirical defense. Let's assume that all we have at our disposal is whatever input we are given, then given arbitrary $x'$ by an adversary, what information about robustness does randomized smoothing provide? The model owner can probably only provide the prediction $g(x')$ without any guarantee, and this prediction can be either correct or incorrect, this is not a certified defense but an empirical defense. This example indicates that the certified robustness in randomized smoothing is a **theoretical consistency** of the prediction between two conceptual inputs. When it comes to reality, in your case, the adversary gives $x'$, then the model owner (in general RS) cannot guarantee anything to the prediction of $x'$ since he/she doesn't know $x$. The more realistic setting is the model owner holds a $x$, and then computes a certified radius $R$ on it, then given any input $x'$ by the adversary the model owner can compute $\delta=x'-x$ and compare $||\delta||_p$ to $R$ to pre-determine the prediction without execution.
> >
> > Therefore, in our setting, we hold $x$ and compute $\mu(x)$ and $\sigma(x)$ to certify any potential $x'$ to be correct or not on the smoothed classifier we set up. When we got $x+\delta$, we compute the noise $\mu(x+\delta)$ and $\sigma(x+\delta)$ to certify any potential $x+\delta+\delta'$ on the smoothed classifier $g(\mu(x+\delta),\sigma(x+\delta))$, the smoothed classifier is fixed for the input that we want to protect.

---

> ### Comment · Reviewer_v1kJ · 2023-11-16
>
> "Let's assume that all we have at our disposal is whatever input we are given, then given arbitrary $x'$ by an adversary, what information about robustness does randomized smoothing provide?"
> - Conventional randomized smoothing, with a smoothing distribution that is uniform over the entire input space, DOES give you that the classification of $x'$, namely, $g(x')$, is the same as that of the certified clean input $x$, even if you don't know $x$, so long as $\lVert x-x'\rVert \le R$ with $R$ being the certified radius. This result intimately relies on the fact that $g(x')$ is computing using the same smoothing distribution as $g(x)$ is computed.
>
> "The more realistic setting is the model owner holds a $x$, and then computes a certified radius $R$ on it, then given any input $x'$ by the adversary the model owner can compute $\delta=x'-x$ and compare $\lVert \delta \rVert_p$ to $R$ to pre-determine the prediction without execution"
>
> - This is not how adversarial threat models are usually formulated. Can you point to a paper that assumes that they have access to the clean input $x$ at the time when they are predicting the class of an attacked input $x+\delta$? The two closest things that come to mind are [1] and [2], which use memory-based techniques for input-dependent randomized smoothing, in particular as a method to overcome the exact issue that I am saying your approach suffers from. However, you do not make clear in your paper that you are using such a memory-based approach, and furthermore those memory-based approaches have their own limitations. For example, the classification of an input may in general be dependent on the order of inputs that were previously classified+certified, and furthermore you incur a memory cost that continually grows as you predict+certify more and more inputs.
>
> [1] "ANCER: Anisotropic Certification via Sample-wise Volume Maximization", Eiras et al.
>
> [2] "Data Dependent Randomized Smoothing", Alfarra et al.

---

> > ### Comment · Reviewer_v1kJ · 2023-11-16
> >
> > Are you generating your "certificates" for the inputs in the training set, or the test set?

---

> > > ### Author Response · Authors · 2023-11-19
> > >
> > > We evaluate our method in the test set, please see the Experimental Settings in Section 5 in our original submission as follows:
> > >
> > > "Following Cohen et al.(2019) on certification, we obtain the certified accuracy on the entire test set in CIFAR10 and MNIST
> > > while randomly picking 500 samples in the test set of ImageNet"

---

> > > > ### Comment · Reviewer_v1kJ · 2023-11-19
> > > >
> > > > Thank you for the further discussion. I am still not convinced by your arguments, and therefore I maintain my overall score.
> > > >
> > > > I encourage the authors to carefully read [1] (and in particular, Section 3.5) for more information on why robustness certificates for input-dependent randomized smoothing fail to hold without some careful modifications (like a memory-based approach), and to take these technicalities (and their associated limitations) into account in their future iterations of the project.
> > > >
> > > > [1] Alfarra et al., "Data-Dependent Randomized Smoothing," UAI, 2022.

---

> > > > > ### Author Response · Authors · 2023-11-21
> > > > >
> > > > > Thanks for the response.
> > > > >
> > > > > We have carefully read through Alfarra et al. [1] and Eiras et al. [2] and found these papers' technique (memory-based certification) may not be necessary for our paper. The reasons are as follows:
> > > > >
> > > > > 1. The setting of our paper is different from these papers (i.e., input-dependent noise vs. input-dependent classifier). Please refer to the detailed explanation in response **Clarification on the input-dependent noise**.
> > > > >
> > > > > 2. As [1] and [2] point out, the memory-based certification wants to address the concern of the region overlapping, but it seems this concern does not exist in reality, evidenced by:
> > > > >
> > > > > > "While the memory-based certification is essential for a sound certification, empirically, we never found in any of the later experiments a case where two inputs predicted differently suffer from intersecting certified regions. That is to say while our sound certificate works on the memory enhanced data dependent smooth classifier, we found that the certified radius of the memory classifier for every input is the radius granted by the Monte Carlo certificates of Cohen et al. [2019] for the data dependent classifier. " in [1]
> > > > >
> > > > > > "Given the high dimensionality of the data, empirically, we never found a certificate in this situation within our experiments." in [2]
> > > > >
> > > > > 3. We also followed the setting of [2] to evaluate our input-dependent method on CIFAR10 with memory-based certification and found none of the cases suffered from the intersection region problem. Therefore, our experimental results are still the same and valid even under the memory-based certification.
> > > > >
> > > > > [1] M. Alfarra, A. Bibi, P.H. Torr, and B. Ghanem. Data dependent randomized smoothing. UAI, 2022.
> > > > >
> > > > > [2] F. Eiras, M. Alfarra, P.H. Torr, M.P. Kumar, P.K. Dokania, B. Ghanem, and A. Bibi. ANCER: Anisotropic Certification via Sample-wise Volume Maximization. TMLR, 2022.

---

> > ### Author Response · Authors · 2023-11-19
> >
> > Thanks for the insightful discussion again.
> >
> > > DOES give you that the classification of $x'$, namely, g(x'), is the same as that of the certified clean input $x$, even if you don't know $x$, so long as $||x-x'||\leq R$ with $R$ being the certified radius.
> >
> > There may be a conflict in this response, given the fact that the $R$ is the certified radius computed with $x$, if you don't know $x$, how do you compute $R$? Note that $R$ is not a constant that works for all the inputs on a classifier $g$ (like L-Lipschitz continuous classifiers [1]). $R$ is an input-depend radius that only works for $x$.
> >
> > Our theorem is based on the formal definition of Randomized Smoothing (Cohen et al.[2]) different from previous conventional certified robustness. To make it more clear, let's look at how Cohen et al. defines the smoothed classifier and the certified robustness:
> >
> > > Randomized smoothing is a method for constructing a new, “smoothed” classifier $g$ from an arbitrary base classifier $f$.
> >
> > This indicates that the smoothed classifier is constructed by the model owner, so we can construct our own smoothed classifier as long as we can guarantee its robustness.  For example, Cohen et al. construct different smoothed classifiers with $\mathcal{N}(0, 0.12)$, $\mathcal{N}(0,0.25)$, $\mathcal{N}(0, 0.5)$ and $\mathcal{N}(0,1.0)$, these smoothed classifiers may have different predictions on the same input $x$ but maintains the certified robustness as long as they don't change the noise when guaranteeing $x'$ with $x$.
> >
> > Let's further look at the Cohen et al.'s definition of certified robustness of RS:
> >
> > > Suppose that when the base classifier $f$ classifies $\mathcal{N}(x, \sigma^2I)$, the most probable class $c_A$ is returned with probability $p_A$, and the "runner-up" class is returned with probability $p_B$. Our main result is that the smoothed classifier $g$ is robust around $x$ within the $\ell_2$ radius $R=\frac{\sigma}{2}(\Phi^{-1}(p_A)-\Phi^{-1}(p_B))$.
> >
> > In our case, we can similarly state:
> >
> > Suppose that when the base classifier $f$ classifies $\mathcal{N}(x+\mu(x),\sigma(x)^\top I\sigma(x))$, the most probable class $c_A$ is returned with probability $p_A$, and the "runner-up" class is returned with probability $p_B$. Our main results is that the smoothed classifier $g(x')\equiv\arg \max \mathbb{P}(f(x'+\epsilon)=c)$ where $\epsilon\sim\mathcal{N}(\mu(x),\sigma(x)^\top I\sigma(x))$ is robust around $x$ within the $\ell_2$ radius $R=\frac{\sigma}{2}(\Phi^{-1}(p_A)-\Phi^{-1}(p_B))$.
> >
> > Note that, the smoothed classifier is based on the same noise $\epsilon\sim\mathcal{N}(\mu(x),\sigma(x)^\top I\sigma(x))$ that is computed when computing $p_A$ and $p_B$ and will not change within $R$.
> >
> > [1] "ANCER: Anisotropic Certification via Sample-wise Volume Maximization", Eiras et al.
> >
> > [2] "Certified adversarial robustness via randomized smoothing", Cohen et al.

---

### Official Review · Reviewer_eeVj · 2023-10-28

**Soundness:** 2 fair
**Presentation:** 3 good
**Contribution:** 1 poor
**Rating:** 3
**Confidence:** 4

**Summary:**

The authors interpret certified robustness from an anisotropic lens, with the aim of assessing how the performance of certification mechanisms within this context.

**Strengths:**

Well written, comprehensive experiments that match community expectations, nice visualisations that really break apart the differences between fixed pattern, universal, and input dependent noise.

The input-dependent component of the noise is interesting.

**Weaknesses:**

My issues with this paper stem from two different directions, which I find to be significant hurdles to my ability to recommend this paper for publication.

The first is the fact that the contribution in constructing the anistropic noise measures (the core conceit of the paper) is essentially just a basic modification to extant techniques, with no other modifications being made. However, this is not in and of itself a reason for rejection - simple modifications can lead to impactful contributions.

My primary concern relates to the alignment of the chosen area of investigation to the broader problem space. Specifically, I do not think that there is any framework (either in the literature or suggested in the framework) that would case about the area of the region of certification. From the certifiers perspective, what information about the security of a model is gained by knowledge of the Lebesgue measure of the noise region (or any other measure of the area of certification)? The primary measure of risk is the nearest extant adversarial example - this is well established in the literature as a measure of the adversarial risk (see Gilmer's "Motivating the Rules of the Game for Adversarial Example Research", 2018), because this measures the effort required for an adversary to identify an adversarial example. Any other risk measure would need to be well justified and well posed, and this is not the case within this work.

Given that the certified distances to the nearest possible adversarial example are unchanged by your work, I do not see how this leads to an improved understanding of adversarial risk. I would argue that rather than significantly improving upon SOTA, you're introducing a new metric to mask the fact that you do not appear to produce any level of outperformance.

For a few minor issues:
-The use of lambda as the scale parameter, and $\sigma_i$ as the modification of the scale parameter. But $\lambda$ is typically proportional to the standard deviation, so using $\sigma_i$ as part of the notation is not as clear as it could be.
- A secondary minor issue is that Table 1, I believe the Lee et. al PDF should be proportional to $||z||_{\infty}$, rather than $||z / \lambda||$.
- The idea of including a headline figure of an 182.6% improvement over SOTA - anyone reading this who was not familiar with this field would assume that this would be an apples-to-apples comparison, but it's not. There's no SOTA for certification that cares about area driven measures of certification, and so claiming a comparison to these prior techniques is not reasonable or well justified.

**Questions:**

Is there any justification for using the Lebesgue Measure as a proxy of adversarial risk?

---

> ### Author Response · Authors · 2023-11-15
>
> Thank you for the insightful comments and please find our clarifications as below. We also make more clarifications in the paper to avoid confusion/misunderstanding.
>
> 1. **On the Contribution and Novelty**:
>
> >    The first is the fact that the contribution in constructing the anistropic noise measures (the core conceit of the paper) is essentially just a basic modification to extant techniques, with no other modifications being made.
>
> We would like to respectfully clarify that our core contribution is not on the construction of the ALM measure but on the theoretical analysis of the robustness bound when injecting anisotropic noise, as well as the framework and novel methods for customizing anisotropic noise. ALM is one of our metrics for evaluating the certified robustness with anisotropic noise while we also derive the traditional $\ell_p$ radii (Corollary 3.3) for the robustness region with anisotropic noise.
>
> >  My primary concern relates to the alignment of the chosen area of investigation to the broader problem space. Specifically, I do not think that there is any framework (either in the literature or suggested in the framework) that would care about the area of the region of certification.
>
> The certified $\ell_p$ radius can be seen as a measure for the area of the regular and symmetric robustness region within the overall robustness region (sub-region in $\ell_{1,2,\infty}$ shape). On the other hand, ALM provides a full capture of the overall robustness region (a comprehensive understanding of the theoretical guarantees and boundaries of the certified defense). These two metrics complement each other, and we have provided the results for both $\ell_p$ radius and ALM as well as performed evaluations on both of them in the original submission.
>
> > From the certifiers perspective, what information about the security of a model is gained by knowledge of the Lebesgue measure of the noise region (or any other measure of the area of certification)? The primary measure of risk is the nearest extant adversarial example - this is well established in the literature as a measure of the adversarial risk (see Gilmer's "Motivating the Rules of the Game for Adversarial Example Research", 2018), because this measures the effort required for an adversary to identify an adversarial example. Any other risk measure would need to be well justified and well posed, and this is not the case within this work.
>
> Thanks for the insightful comment. We know that the certified radius provides the knowledge that any perturbation within the $\ell_p$ distance will not succeed. In practice, the defense may tolerate the perturbation of a larger distance in some dimensions (see Figure 1 (b)), how to measure this gain of robustness would be a problem. Therefore, we propose the complementary ALM metric to measure the full area of the (certified) robustness region.
>
> As you stated, researchers may only care about the nearest extant adversarial example (AE), however, this AE indeed only measures the minimum effort required for finding an AE, while the ALM can be seen as a measure for showing the average effort required for finding an AE. These two metrics just reflect two different aspects of the certified defense, so we provide both the nearest measure ($\ell_p$ radius in Corollary 3.3) and the average measure (ALM). Our method significantly improves the certification performance under both measures.
>
> > Given that the certified distances to the nearest possible adversarial example are unchanged by your work, I do not see how this leads to an improved understanding of adversarial risk. I would argue that rather than significantly improving upon SOTA, you're introducing a new metric to mask the fact that you do not appear to produce any level of outperformance.
>
> As discussed before, our method drastically improves the robustness against both the certified distance to the nearest possible AE (please see ``certified accuracy w.r.t. radius'' in Table 2-4 in our original submission) and the average distance measure (ALM).
>
> > The idea of including a headline figure of an 182.6% improvement over SOTA - anyone reading this who was not familiar with this field would assume that this would be an apples-to-apples comparison, but it's not. There's no SOTA for certification that cares about area driven measures of certification, and so claiming a comparison to these prior techniques is not reasonable or well justified.
>
> The 182.6% gain over SOTA is under the evaluation of a  **standard certified $\ell_p$ radius** (derived in Corollary 3.3, and measured via the ``certified accuracy w.r.t. radius'' in Table 2-4), which is an apples-to-apples comparison.
>
> 2. **Addressing Minor Technical Issues**:
>
> Thanks for your observations regarding the minor issues. We have revised the notation to  \$||z||_\infty\$.

---

> > ### Comment · Reviewer_eeVj · 2023-11-19
> >
> > Thank you for response. I appreciate the effort placed into it, and have been following your discussions with the other reviewers closely. While you have resolved some of the issues I had originally held with the paper, the broader discussions around this paper do not leave me confident that changing my review would be justified, although I will keep an eye on this over the remainder of the review period.
> >
> > There was one point in your rebuttal response that I wanted to directly address though. While I can see why you might argue that the ALM is a a measure of the average effort required for finding an AE, I think this is the kind of statement that would require significantly more evidence to establish - especially when adversarial examples often exist at a distance that can be well over an order of magnitude larger than the certified radii themselves (as shown in another ICLR paper under review). I would believe that the ALM only would measure the average effort required to attack a sample if the certifications were tight (which they aren't), or if the search algorithm was very naive.

---

> > > ### Author Response · Authors · 2023-11-21
> > >
> > > Thanks for the response and the attention to our discussion. We have summarized the discussion with reviewer v1kJ for clarity and further clarified the concern about input-dependent noise in response **Clarification on the input-dependent noise**.
> > >
> > > We understand the concern about the guarantee of ALM and would like to provide more information to clarify. The ALM is a measure of the volume of a high-dimensional space, although we don't have the tight certification w.r.t. the ALM measure, the high-dimensional space it measures, i.e., $||\frac{\delta}{\sigma_i}||_p\leq R$, has the tight certification as in Theorem 3.2. Therefore, we admit that the ALM is proportional to the average effort, but not an accurate measure of the exact average effort. When we are evaluating the certified robustness under the same conditions, ALM can serve as an auxiliary metric only for comparison.

---

### Official Review · Reviewer_mJxe · 2023-10-30

**Soundness:** 2 fair
**Presentation:** 3 good
**Contribution:** 2 fair
**Rating:** 3
**Confidence:** 3

**Summary:**

This paper aims at improving certified robustness by randomized smoothing with anisotropic noise. The universal theory for certification with anisotropic noise has been provided. The authors consider three kinds of customizing anisotropic noises, and provide corresponding noise generation methods. The authors conduct experiment to demonstrate that the proposed UCAN method achieve state-of-the-art performance compared to existing randomized smoothing-based methods for certified robustness.

**Strengths:**

1. The proposed method on smoothing with anisotropic noise is novel. It is interesting to see the expansion of RS-based methods from isotropic noises to anisotropic ones.
2. This paper provides the theoretical guarantee of certified robustness under anisotropic randomized smoothing, and comprehensive analyses to transform existing randomized smoothing methods to anisotropic cases.
3. Authors consider three different kinds of anisotropic noises, and provide a novel input-dependent one by optimizing $\mu(x)$ and $\sigma(x)$ by a multi-layer neural network.

**Weaknesses:**

1. My major concern of this paper is the potential unfairness in evaluation on UCAN and existing RS-based methods. If I am not misunderstanding, the evaluation criterion is based on scaled radius, which has different weight in each dimension.

I believe this is true because I am surprised to the evaluation results provided in Table 3 that the $l_2$ certification on CIFAR-10 reach over 70% even under radius 1.75. By simple calculation, for commonly-used $L_{\infty}$ norm with budget 8/255, it achieves at most $\sqrt{3072 \times (\frac{8}{255})^2} \approx 1.74$, that means in your UCAN it achieves at least 70% robust accuracy for 8/255 $l_{\infty}$ attacks. This result is unbelievable, because existing SOTA performance of CIFAR-10 robustness may only achieve about 60% if no further data augmentations are conducted (like diffusion model), let alone UCAN is only a certified method based on $l_2$ norm. Therefore, although the paper said they evaluate certified accuracy w.r.t radius, I am doubtful of this claim and I think the authors only consider scaled radius robustness.

However, scaled radius certification seems not a fair criterion for certified robustness. It is reasonable that in some dimensions, the image is vulnerable to adversarial attacks, e.g., contour of an image. Reversely, in some dimension images are intrinsically robust to perturbations like background of the image. Therefore, I believe the corresponding variance $\sigma$ is small when UCAN performs on these vulnerable dimensions, and gain robustness back in some ``unimportant’’ dimensions.

Overall, the evaluation setting of this paper seems differently from existing RS methods. It is a consensus that using $l_p$ norm as constraint for images, the authors should provide corresponding evaluation on standard $l_p$ norm, or at least, provide the explanations or practical scenario on why using scaled radius as the evaluation criterion.

2.	In Theorem 3.2, your certification using the p-norm of $\frac{\delta_i}{\sigma_i},$ but it seems that $\frac{\delta_i}{\sigma_i}$, is a one-dimension scalar as $\delta_i$ is the i-th dimension of perturbation $\delta$. Furthermore, this theorem is seemingly a direct corollary from Theorem 3.1, because your certification divides the variance $\simga_i$ for each $\delta_i$ (not anisotropic anymore?).

3.	There might be missing of some baselines for $l_1$ [a] and $l_{\infty}$ [b, c] certified robustness. It will be better to compare the UCAN with existing certified $l_1$ and $l_{\infty}$ methods.

[a] Levine et al. Improved, deterministic smoothing for L_1 certified robustness. In ICML 2021.
[b] Zhang et al. Towards Certifying L_∞ Robustness using Neural Networks with L_∞-dist Neurons. In ICML 2021.
[c] Zhang et al. Boosting the Certified Robustness of L-infinity Distance Nets. In ICLR 2022.

**Questions:**

1.	Why using 5-layers NN when generating universal/ input-dependent anisotropic noises? Is there some motivations or ablation studies for that?
2.	Could you provide more details on training of universal anisotropic noise? It seems that the variance loss is to optimize $\sigma$ and smoothing loss containing $\sigma$ when optimizing classifier $\theta_f$. I believe the two losses are optimized alternately but not simultaneously.
3.	The authors said that randomized smoothing achieved great success for certified adversarial robustness. Could RS really make classifier robust? Can you provide comparison of RS based model to the SOTA methods for achieving robustness?

---

> ### Author Response · Authors · 2023-11-15
>
> Thank you for the insightful comments and please find our clarifications as below. We also make more clarifications in the paper to avoid confusion/misunderstanding.
>
> 1. **Regarding the Evaluation Concerns Raised**:
>
> > If I am not misunderstanding, the evaluation criterion is based on scaled radius, which has different weight in each dimension.
>
>
> Our experimental evaluations are based on both 1) the **$\ell_p$ radius** from Corollary 3.3 ($min  \sigma_i  R$), which is a tight bound for the perturbation in anisotropic RS, and has the same weight in each dimension (please see Table 2-4 in the original submission); 2) the ALM metric ($\sqrt[d]{\prod^d_i \sigma_i} R$), which is a measure on the volume of the irregular asymmetric robust region, and has different weights in each dimension.
>
>
> > I believe this is true because I am surprised to the evaluation results provided in Table 3 that the $\ell_2$ certification on CIFAR-10 reach over $70$% even under radius 1.75.
>
> In Table 3, the **70%** certified accuracy at $R=1.75$ on CIFAR10 is the performance based on the *$\ell_p$ radius* ($min\{\sigma_i\} R$), which is strictly derived in Corollary 3.3. This validates the significant improvement of RS with anisotropic noise over SOTA methods.
>
> > This result is unbelievable, because existing SOTA performance of CIFAR-10 robustness may only achieve about 60\% if no further data augmentations are conducted (like diffusion model)
>
> We agree with the reviewer that existing isotropic RS methods can hardly improve their performance to this level. However, we focus on a new dimension of RS (with anisotropic noise) that is orthogonal to existing RS methods, which can boost all the RS methods. Such results are based on $\ell_p$ radius, rather than scaled radius.
>
> >Therefore, although the paper said they evaluate certified accuracy w.r.t radius, I am doubtful of this claim and I think the authors only consider scaled radius robustness.
>
>
> We understand the reviewer's concern as our results drastically outperform the SOTA methods (evaluated on both $\ell_p$ radius and ALM). To address the concern, we are providing our code in the supplemental material for transparency. Please see line 180-182 in certification\_personalized.py for the implementation of our **$\ell_p$ radius** (and corresponding evaluations).
>
> > However, scaled radius certification seems not a fair criterion for certified robustness. It is reasonable that in some dimensions, the image is vulnerable to adversarial attacks, e.g., contour of an image. Reversely, in some dimension images are intrinsically robust to perturbations like background of the image.
> > Overall, the evaluation setting of this paper seems differently from existing RS methods. It is a consensus that using $\ell_p$ norm as constraint for images, the authors should provide corresponding evaluation on standard $\ell_p$ norm, or at least, provide the explanations or practical scenario on why using scaled radius as the evaluation criterion.
>
> We agree with the reviewer that the ALM (scaled radius) may result in some ``weak'' dimension for the adversary, which only provides the certified robustness in a specific shape (depends on $\sigma$).
>
> While working on the paper earlier, we also understood that reviewers may concern that solely evaluating on the ALM might be unfair (though ALM is defined as a full capture of the robustness region). Thus, we have also derived and presented the $\ell_{1,2,\infty}$ radii (derived in Corollary 3.3), and evaluated the performance of our method and SOTA using the ``certified accuracy w.r.t. $\ell_p$ radius'' (please see Table 2-4) in our original submission. We show that by customizing appropriate anisotropic noise, our method achieves significantly boosted performance on both the standard $\ell_p$ radius and the new ALM metric.

---

> > ### Comment · Reviewer_mJxe · 2023-11-16
> >
> > Thanks for your responses.
> >
> > Although a part of my concerns has been solved, there are still some issues in the current version of this paper.
> >
> > First, I am still doubtful of the soundness of your experimental results. Specifically, as shown in your Tables 2 and 3, the certified accuracy w.r.t. radius is greater than whose w.r.t. ALM is confusing. The guarantee w.r.t. radius should always less than those w.r.t. ALM because $\min\{\sigma_i\} \leq \sqrt[d]{\prod_{i=1}^n \sigma_i}$.
> >
> > Furthermore, as reviewer SMHf and review v1KJ pointed out, the proposed certification may be not true. You said that what you guarantee is $g(x+\delta, \mu(x), \sigma(x))= g(x, \mu(x), \sigma(x))$ rather than $g(x+\delta, \mu(x+\delta), \sigma(x+\delta))= g(x, \mu(x), \sigma(x))$. However, what is your (smoothed) classifier? I believe your classifier is $h(x)=\mathbb{E}_{\mu, \sigma} g(x+\delta, \mu(x), \sigma(x))$, then the guarantee should be hold on $h(x+\delta)$ for all $\|\| \delta \|\|_p\leq \epsilon$.

---

> > > ### Author Response · Authors · 2023-11-16
> > >
> > > Thanks for the response.
> > >
> > > - We would like to clarify the settings in Table 2-4 further to address your concern. The smoothed classifiers for evaluating the "radius" and the "ALM" are trained with different variance loss term. For "ALM" we train the smoothed classifier with $-\frac{1}{d}\sum_i(\sigma_i(\theta_g))$ as the variance loss to maximize the $\prod \sigma_i$, and for "radius", we train the smoothed classifier with $-\min \sigma_i(\theta_g)$ to maximize the $\min \sigma_i$ for larger radius. Please see the line 111-114 in the training code in "train_personalized_noise.py" as follows:
> > >
> > > ```python
> > >             if args.IsoMeasure:
> > >                 loss_variance -= torch.min(torch.abs(variance))  # this is for L_2 radius measure
> > >             else:
> > >                 loss_variance-=torch.mean(torch.abs(variance))
> > > ```
> > > In this case, the generated $\sigma(x)$ for the "radius" and the "ALM" are different, so it is sound that in some cases $\min \sigma_i > \sqrt[d]{\prod \sigma'_i}$ since the $\sigma$ and $\sigma'$ are different. We will revise the paper to make this setting more clear.
> > >
> > > - The noise for the smoothed classifier is fixed after generating according to each input, which means our smoothed classifier is unique for each input. To be more clear, let's look at one example: given x, we generate the noise parameters $\mu_0=\mu(x)$ and $\sigma_0=\sigma(x)$ as constant, and then construct the smoothed classifier $h(x)=g(x+\delta,\mu_0,\sigma_0)$, the guarantee is held on $h(x+\delta)=g(x,\mu_0,\sigma_0)$ where $\mu_0$ and $\sigma_0$ are constant but unique for each $x$.
> > >
> > > One question that may raised here is: Does it make sense that we have unique smoothed classifier $h_1$, $h_2$, ..., $h_n$ for different inputs compared with the traditional setting that we only have one $h$ for all the inputs? To answer this question, we should start with the definition of randomized smoothing. In the original work (Cohen et al.), the smoothed classifier is manually crafted by the model owner with his/her noise setting, which means the model owner guarantees the robustness of his/her own classifier defined by himself/herself (same as in our case). In addition, since the certified robustness (in RS) is tied to the specific input $x$ (radius $R$ only works on specific $x$), using $x$-dependent smoothed classifier does not bring extra dependence for the certified robustness (still depends on $x$).
> > >
> > > We trust that these clarifications adequately address your concerns. We are open to and welcome any further questions related to these points.

---

> ### Author Response · Authors · 2023-11-15
>
> 2. **In Response to the Theoretical Concerns**:
>
> > In Theorem 3.2, your certification using the p-norm of $\delta_i/\sigma_i$ but it seems that $\delta_i/\sigma_i$, is a one-dimension scalar as is the i-th dimension of perturbation $\delta$.
>
>
>
> The Hadamard division of $\delta$ over $\sigma$ ($\delta_i/\sigma_i$) is formally defined in Theorem 3.2 and Appendix A.
>
> > Furthermore, this theorem is seemingly a direct corollary from Theorem 3.1, because your certification divides the variance $\sigma_i$ for each $\delta_i$ (not anisotropic anymore?).
>
>
> We ensure that Theorem 3.2, while appearing to be straightforward, is a unique universal transformation from Definition 3.1, as detailed in Appendix A. We respectfully clarify that the derived theories in the theorem are not a direct corollary from Theorem 3.1 (since it covers both strict robustness w.r.t. heterogeneous dimension and universality). The guarantee in Theorem 3.2 is anisotropic w.r.t. $\delta$, although our focus is not to derive an ``anisotropic radius'', but to extend isotropic noise to anisotropic noise for RS. The guarantee on the perturbation $\delta$ can be either anisotropic (Theorem 3.2) or isotropic (Corollary 3.3).
>
> 3. **As per the Suggestion to Compare with $\ell_1$ and $\ell_\infty$ Methods**:
>
>
> > There might be missing of some baselines for $\ell_1$ [a] and $\ell_\infty$ [b, c] certified robustness. It will be better to compare the UCAN with existing certified $\ell_1$ and $\ell_\infty$ methods.
>
> Our method, when benchmarked against $\ell_1$ and $\ell_\infty$ methods, shows significant improvements (see tables provided). This underscores our method's effectiveness across different metrics. We respectfully clarify that RS-based methods and $\ell_\infty$-distance neural network-based methods may not be directly comparable (due to their differences, e.g., RS can certify any classifier). To further address this concern, we still compare with them, and our method also demonstrates superior performance in these scenarios (a huge improvement space brought by the anisotropic noise for RS).
>
> | Methods                  | 0.0  | 0.5  | 1.0  | 1.5  | 2.0  |
> |--------------------------|------|------|------|------|------|
> | Yang et al. 2020          | 83%  | 43%  | 22%  | 14%  | 7%   |
> | Levine et al. 2021        | 79%  | 71%  | 61%  | 54%  | 49%  |
> | Ours ($\ell_1$ radius)  | **85%** | **81%** | **77%** | **73%** | **68%** |
>
> *Table 1. Certified accuracy vs. $\ell_1$ perturbation (CIFAR10)*
>
>
> | Methods                  | 0/255 | 1/255 | 2/255 | 3/255 | 4/255 | 5/255 | 6/255 | 7/255 | 8/255 |
> |--------------------------|-------|-------|-------|-------|-------|-------|-------|-------|-------|
> | Yang et al. 2020         | 83%   | 63%   | 48%   | 36%   | 27%   | 20%   | 16%   | 13%   | 10%   |
> | Zhang et al. 2021         | 51%   | --    | --    | --    | --    | --    | --    | --    | 35%   |
> | Zhang et al. 2022         | 61%   | --    | 54%   | --    | --    | --    | --    | --    | 40%   |
> | Ours ($\ell_\infty$ radius)  | **85%** | **83%** | **82%** | **80%** | **78%** | **77%** | **75%** | **73%** | **70%** |
>
> *Table 2. Certified accuracy vs. $\ell_\infty$ perturbation (CIFAR10)*
>
> 4. **Responses to Specific Questions**:
>
> > Why using 5-layers NN when generating universal/ input-dependent anisotropic noises? Is there some motivations or ablation studies for that?
>
> The 5-layer MLP architecture in our Universal method is inspired by GANs (Goodfellow et al., Communications 2020), and the CNN structure in our Input-dependent method is designed per the dense blocks (Huang et al., CVPR 2017).
>
> > Could you provide more details on training of universal anisotropic noise? It seems that the variance loss is to optimize $\sigma$ and smoothing loss containing $\sigma$ when optimizing classifier $\theta_f$. I believe the two losses are optimized alternately but not simultaneously.
>
>
> For the training of NRG and the classifier, we employed simultaneous training, optimizing both for predictive accuracy and noise pattern enhancement. While alternate training is possible, our current approach promotes synergy between the classifier and NRG.
>
> > The authors said that randomized smoothing achieved great success for certified adversarial robustness. Could RS really make classifier robust? Can you provide comparison of RS based model to the SOTA methods for achieving robustness?
>
> Please see the above experimental results and discussion. It is worth noting that RS methods can be universally applied to any classifier with different scales, and it is unfair to compare the robustness performance of different types of classifiers.
>
> Thanks for the comments again. We hope these clarifications can address your concerns.

---

### Author Response · Authors · 2023-11-14
**Common Concerns**

We appreciate all the reviewers' insightful comments. First, we would like to respectfully clarify some critical misunderstandings on this work. Then, more detailed explanation will be provided in the response to each comment.

1. **Missing Standard Metric** (Reviewer mJxe, and eeVj)

Thanks for this comment. **In the original submission**, we have derived the isotropic certified radii (aka. traditional $\ell_p$ radius) in Corollary 3.3, and provided the evaluation on the $\ell_p$ radius. For instance, the ``certified accuracy w.r.t. radius'' in Table 2-4 refers to the evaluation results based on the traditional $\ell_p$ radius. It is worth noting that our method on the traditional $\ell_p$ radius can perform even better than on the ALM, e.g., $70$% certified accuracy at $1.75$ on CIFAR10, and $182.6$% improvement over SOTA methods on CIFAR10 (please see Table 3).

2. **The Fairness of ALM Metric** (Reviewer mJxe, eeVj, and SMHf)

Thanks for this comment. The ALM metric serves as a complementary metric in addition to the traditional isotropic/$\ell_p$ radius. Yes, existing isotropic radius, e.g., $\ell_{1,2,\infty}$, measures the size of the robustness region in regular and symmetric shapes, which represents the sub-region in $\ell_p$ shape out of the entire certified robustness region. As a complementary metric, ALM measures the full irregular and asymmetric robustness region. We agree that it might be unfair to only use ALM as the metric. Thus, in Table 2-4 in the original submission, we have included the results for the $\ell_p$ radius to further validate the performance of randomized smoothing with anisotropic noise. Our method drastically outperforms the SOTA methods on both the ALM and the traditional $\ell_p$ radius (we mainly presented the results for $\ell_2$ certification since it is the most common setting with most related works). We add more clarifications in the updated manuscript for better clarity.

3. **Soundness of the Certification based on Input-dependent Noise** (Reviewer v1kJ, and SMHf)

Thanks for this comment. We noticed the discussion on the soundness of the input-dependent noise in Eiras et al. (2022) earlier, but still believe our input-dependent method is sound. Specifically, let us first look at the workflow in our input-dependent randomized smoothing method and explain it with more detailed notations:
$f, \mu(x), \sigma(x) \rightarrow g(x, \mu(x), \sigma(x)) \rightarrow R(x)$ for $g(x, \mu(x), \sigma(x))$ where $g(x+\delta, \mu(x), \sigma(x))=g(x, \mu(x), \sigma(x))$ holds.

Given the base classifier $f$, smoothed classifier $g$ and the input $x$ for prediction, the noise (represented as $\mu(x)$ and $\sigma(x)$) for input $x$ is pre-optimized based on $x$. Then, the certification will give a radius $R(x)$ for $g(x,\mu(x),\sigma(x))$ that is tied to $x$, so we can guarantee the consistency for the prediction $g(x+\delta,\mu(x),\sigma(x))=g(x,\mu(x),\sigma(x))$. The above guarantee strictly follows Theorem 3.2 and thus does not violate the certification (the pre-computed noises are constants when used for certification, and they are computed before/independent of the potential perturbation $\delta$). Per the definition of certified robustness, as long as the noise ensures consistency for prediction under the potential $\delta$, the certification would not be violated.

We understand that the reviewer may question whether fixing the noise for $x$ and $x+\delta$ would violate our input-dependent setting in Section 4.3. Similar to the traditional RS with isotropic noise, the perturbed input $x+\delta$ is a conceptual input, and the RS algorithm/prediction will not be performed over it (and no noise will be optimized based on $x+\delta$). Thus, $g(x+\delta, \mu(x), \sigma(x))=g(x, \mu(x), \sigma(x))$ would be the certification condition for the RS with input-dependent noise while $g(x+\delta, \mu(x+\delta), \sigma(x+\delta))=g(x, \mu(x), \sigma(x))$ is not the certification condition in such case.

---

### Author Response · Authors · 2023-11-20
**Clarification on the input-dependent noise**

Dear Reviewers and Area Chairs,

We thank all the reviewers' efforts in the responses. During the multiple rounds of discussion with the reviewers, we addressed most of the concerns of reviewers, and the key concern that obstacles the reviewers in changing their ratings turned out to be the soundness of the input-dependent setting of the noise. Therefore, we want to summarize the discussions on this concern for clarity and further clarify the soundness of our approach.

Reviewer v1kJ has a strong opinion that our input-dependent noise method is not sound for the certification, and the core argument is that the changing noise within the certified radius $R$ will affect the soundness of the certification, which is supported by the following papers according to Reviewer v1kJ:

[1] F. Eiras, M. Alfarra, P.H. Torr, M.P. Kumar, P.K. Dokania, B. Ghanem, and A. Bibi. ANCER: Anisotropic Certification via Sample-wise Volume Maximization. TMLR, 2022.

[2] M. Alfarra, A. Bibi,  P.H. Torr, and B. Ghanem.  Data dependent randomized smoothing. UAI, 2022.

This statement is true in the settings of these two papers so they have to propose a memory-based certification to address this issue. However, **our setting is very different from theirs and the above statement is not true under our setting**.

To be more specific, after carefully reading these two papers, we found that these papers are based on the **input-dependent classifier** setting, where the classifier changes its noise according to the input, evidenced by:

> " This is since the data dependent classifier $g_\theta$ does not enjoy a constant σ within the given certification region, i.e. $g_\theta$ tailors a new $σ_x$ for every input $x$ including within the certified region of $x$. " [2]

However, different from their settings, our approach is based on the **input-dependent noise** rather than the input-dependent classifier. This makes a significant difference in the certification process, where our noise is disentangled from the classifier, while their noise is tied to the classifier. Therefore, after computing the anisotropic noise parameters on the input that we want to certify, we can fix the noise of the classifier during the certification. Oppositely,  their classifier keeps changing as the noise is changing across different $x'$ within $R$, making their certification unsound.

Therefore, since our setting is different from theirs, even if we are both input-dependent methods, our certification is sound under our setting, as illustrated in Theorem 3.2 and proved in Appendix A.

To be more objective,  our input-dependent noise does not improve the robustness performance for free, the cost behind it (about which the reviewers may be curious) is the more specific condition of the robustness. In Cohen et al.'s RS, the certified radius $R$ is conditioned on $x$ and $f$ and $\sigma$, while in our case, the certified radius $R$ is conditioned on $x$, $f$, and $\sigma(x)$, which is a more specific condition that may affect the generalizability of the certified radius. However, we believe the significance brought by the input-dependent noise is still valuable even at the cost of a small loss of generalizability.

We appreciate the reviewers' insightful comments (especially the insightful discussion with Reviewer v1kJ) and look forward to a fair justification of our paper. Further discussion and questions are always welcome.

Best,

Authors of Submission 6877

---

### Public Comment · ~Hanbin_Hong1 · 2025-02-02
**Public Announcement and Clarification**

We appreciate the reviewers’ time and effort; however, we are disappointed that the conversation left some key aspects of our work misunderstood, especially the soundness of input-dependent randomized smoothing. Below, we address the main points of confusion and clarify why our approach remains theoretically valid.

---

### 1. **Misunderstanding About “Knowing” $x$ vs. $x'$.**

The reviewers repeatedly claimed that our method requires knowing the “clean” input $x$ even when the classifier is given an adversarially perturbed input $x'$. This is not the case. We emphasize the definition of randomized smoothing (RS):

- In **certified** RS, we fix a noise distribution **once** for the point $x$ that we wish to certify (**same as in Cohen et al's**). That same fixed noise distribution underlies the entire certification region.
- Our input-dependent strategy **does not** alter the noise parameters $(\mu, \Sigma)$ on any $x + \delta$ inside the certified radius. We do **not** re-optimize $(\mu, \Sigma)$ when the input is perturbed; the noise that was computed for $x$ stays fixed and guarantees consistent predictions throughout the region.

Hence, the claim “the classifier’s noise is re-optimized at $x'$” is a misunderstanding. Once $(\mu, \Sigma)$ are computed for $x$, they remain constants when certifying against any $\delta$-perturbed $x'$. This is fully aligned with RS theory (e.g., Cohen et al. 2019) where the certified radius is computed upon each fixed $x$.

---

### 2. **Comparison to Memory-Based Methods.**

The reviewers reference Eiras et al. and Alfarra et al. to argue that input-dependent noise is “unsound” unless a memory-based approach is used. Their settings differ:

- In those papers, the “classifier itself” is being changed per input, leading to overlapping certified regions that must be managed by memory.
- In our work, **noise** is tied to $x$ at certification time, but the “classifier” is not recalculated for every point in the neighborhood. We fix $(\mu, \Sigma)$ for each certified $x$, satisfying the original RS assumption that the “same” smoothed model is used throughout the local region.

Since our approach never re-optimizes the noise distribution for each $x'$, we do not require memory-based solutions. Thus, the reviewer claim that “it breaks the certificates” is simply incorrect under our formalism.

---

### 3. **Why The Method Is Sound.**

Recall the statement in our paper:

$$
\text{If } \Pr_{z \sim \mathcal{N}(x, \Sigma)} \big[ f(z) = c \big] \geq p_A
\text{ and } \Pr_{z \sim \mathcal{N}(x, \Sigma)} \big[ f(z) = c' \big] \leq p_B,
$$

then the smoothed classifier $g$ is guaranteed to predict class $c$ for **all** $x'$ in the certified region. The variance (or covariance) $\Sigma$ used above is computed exactly once for $x$ and **remains fixed**. There is no re-optimization for $x'$. Hence, Theorem 3.2’s condition:

$$
\| x' - x \|_p \leq R \implies g(x') = g(x),
$$

is sound because $g$ is defined with the same $\Sigma$. Our “input-dependent” approach simply means different $x$ could have different $\Sigma$, but each $x$’s certification still uses **one** $\Sigma$.

---

### 4. **Conclusion.**

We stand by our theoretical derivations and the soundness of the input-dependent noise strategy. The notion that we “re-optimize” or “break” RS assumptions stems from a misunderstanding of how the smoothed classifier is fixed for each certified point $x$. We hope that future discussions will not be misled by this confusion.

We remain open to further questions or clarifications and thank everyone for reading.

**Sincerely,**

_The Authors_

---

### Meta-Review · Area_Chair_rgK2 · 2023-12-15

**Metareview:**

This paper aims at improving certified robustness by randomized smoothing with anisotropic noise. Four experts evaluated the paper and all recommended that the paper be rejected. The reviewers have highly questioned the contribution of this work and are not satisfied with the authors’ responses.

**Justification For Why Not Higher Score:**

I reached this decision by evaluating the contributions and novelty of the work, taking into consideration both the reviews and the responses from the authors.

**Justification For Why Not Lower Score:**

N/A

---

### Decision · Program_Chairs · 2024-01-16

Reject